# VEM-GCN:
# Topology Optimization with Variational EM for Graph Convolutional Networks

## Abstract

Over-smoothing has emerged as a severe problem for node classification with graph convolutional networks (GCNs). In the view of message passing, the over-smoothing issue is caused by the observed noisy graph topology that would propagate information along inter-class edges, and consequently, over-mix the features of nodes in different classes. In this paper, we propose a novel architecture, namely VEM-GCN, to address this problem by employing the variational EM algorithm to jointly optimize the graph topology and learn desirable node representations for classification. Specifically, variational EM approaches a latent adjacency matrix parameterized by the assortative-constrained stochastic block model (SBM) to enhance intra-class connection and suppress inter-class interaction of the observed noisy graph. In the variational E-step, graph topology is optimized by approximating the posterior probability distribution of the latent adjacency matrix with a neural network learned from node embeddings. In the M-step, node representations are learned using the graph convolutional network based on the refined graph topology for the downstream task of classification. VEM-GCN is demonstrated to outperform existing strategies for tackling over-smoothing and optimizing graph topology in node classification on seven benchmark datasets.

## 1 Introduction

Complex graph-structured data are ubiquitous in the real world, ranging from social networks to chemical molecules. Inspired by the remarkable performance of convolutional neural networks (CNNs) in processing data with regular grid structures (e.g., images), a myriad of studies on GCNs have emerged to execute "convolution" in the graph domain (Niepert et al., 2016; Kipf & Welling, 2017; Gilmer et al., 2017; Hamilton et al., 2017; Monti et al., 2017; Gao et al., 2018). Many of these approaches follow a neighborhood aggregation mechanism (a.k.a., message passing scheme) that updates the representation of each node by iteratively aggregating the transformed messages sent from its neighboring nodes. Commencing with the pioneering works (Kipf & Welling, 2017; Gilmer et al., 2017), numerous strategies have been developed to improve the vanilla message passing scheme such as introducing self-attention mechanism (Veličković et al., 2018; Zhang et al., 2020), incorporating local structural information (Zhang et al., 2020; Jin et al., 2019; Ye et al., 2020), and leveraging the link attributes (Gong & Cheng, 2019; Li et al., 2019; Jiang et al., 2019).

Despite significant success in many fundamental tasks of graph-based machine learning, message passing-based GCNs almost all process the observed graph structure as ground truth and might suffer from the over-smoothing problem (Li et al., 2018), which would seriously affect the node classification performance. Given the observed noisy graph topology (i.e., excessive inter-class edges are linked while many intra-class edges are missing), when multiple message passing layers are stacked to enlarge the receptive field (the maximum hop of neighborhoods), features of neighboring nodes in different classes would be dominant in message passing. Thus, node representations would be corrupted by the harmful noise and affect the discrimination of graph nodes. The over-smoothing phenomenon in GCNs has already been studied from different aspects. Li et al. (2018) first interpreted over-smoothing from the perspective of Laplacian smoothing, while Xu et al. (2018) and Klicpera et al. (2019a) associated it with the limit distribution of random walk. Furthermore, Chen et al. (2020a) developed quantitative metrics to measure the over-smoothness from the topological

view. They argued that the key factor leading to over-smoothing is the noise passing between nodes of different categories and the classification performance of GCNs is positively correlated with the proportion of intra-class node pairs in all edges.

In this paper, we propose VEM-GCN, a novel architecture to address the over-smoothing problem with topology optimization for uncertain graphs. Considering that a "clearer" graph with more intra-class edges and fewer inter-class edges would improve the node classification performance of GCNs (Yang et al., 2019; Chen et al., 2020a), VEM-GCN approaches a latent adjacency matrix parameterized by the assortative-constrained stochastic block model (SBM) where nodes share the same label are linked and inter-class edges should be cut off. To jointly refine the latent graph structure and learn desirable node representations for classification, variational EM algorithm (Neal & Hinton, 1998) is adopted to optimize the evidence lower bound (ELBO) of the likelihood function. In the inference procedure (E-step), graph topology is optimized by approximating the posterior probability distribution of the latent adjacency matrix with a neural network learned from node embeddings. In the learning procedure (M-step), a conventional GCN is trained to maximize the log-likelihood of the observed node labels based on the learned latent graph structure. The E-step and M-step optimize the graph topology and improve the classification of unlabeled nodes in an alternating fashion.

The proposed VEM-GCN architecture is flexible and general. In the E-step, the neural network can support arbitrary desirable node embeddings generated by algorithms such as node2vec (Grover & Leskovec, 2016), struc2vec (Ribeiro et al., 2017), and GCNs, or the raw node attributes. The GCN in the M-step can also be substituted with arbitrary graph models. Furthermore, recent strategies for relieving the over-smoothing issue, i.e., AdaEdge (Chen et al., 2020a) and DropEdge (Rong et al., 2020), are shown to be the specific cases of VEM-GCN under certain conditions. For empirical evaluation, we conduct extensive experiments on seven benchmarks for node classification, including four citation networks, two Amazon co-purchase graphs, and one Microsoft Academic graph. Experimental results demonstrate the effectiveness of the proposed VEM-GCN architecture in optimizing graph topology and mitigating the over-smoothing problem for GCNs.

## 2 BACKGROUND AND RELATED WORKS

**Problem Setting.** This paper focuses on the task of graph-based transductive node classification. A simple attributed graph is defined as a tuple $\mathcal{G}_{\text{obs}} = (\mathcal{V}, \mathbf{A}_{\text{obs}}, \mathbf{X})$, where $\mathcal{V} = \{v_i\}_{i=1}^N$ is the node set, $\mathbf{A}_{\text{obs}} = \left[ a_{ij}^{\text{obs}} \right] \in \{0, 1\}^{N \times N}$ is the observed adjacency matrix, and $\mathbf{X} \in \mathbb{R}^{N \times f}$ represents the collection of attributes with each row corresponding to the features of an individual node. Given the labels $\mathbf{Y}_l = [y_{ic}] \in \{0, 1\}^{|\mathcal{V}_l| \times C}$ for a subset of graph nodes $\mathcal{V}_l \subset \mathcal{V}$ assigned to $C$ classes, the task is to infer the classes $\mathbf{Y}_u = [y_{jc}] \in \{0, 1\}^{|\mathcal{V}_u| \times C}$ for the unlabeled nodes $\mathcal{V}_u = \mathcal{V} \backslash \mathcal{V}_l$ based on $\mathcal{G}_{\text{obs}}$.

**Graph Convolutional Networks (GCNs).** The core of most GCNs is message passing scheme, where each node updates its representation by iteratively aggregating features from its neighborhoods. Denote with $\mathbf{W}^{(l)}$ the learnable weights in the $l$-th layer, $\mathcal{N}(i)$ the set of neighboring node indices for node $v_i$, and $\sigma(\cdot)$ the nonlinear activation function. A basic message passing layer takes the following form:

$$\mathbf{h}_i^{(l+1)} = \sigma \left( \sum\nolimits_{j \in \mathcal{N}(i) \cup \{i\}} \alpha_{ij}^{(l)} \mathbf{W}^{(l)} \mathbf{h}_j^{(l)} \right). \tag{1}$$

Here, $\mathbf{h}_j^{(l)}$ is the input features of node $v_j$ in the $l$-th layer, $\mathbf{W}^{(l)} \mathbf{h}_j^{(l)}$ is the corresponding transformed message, and $\alpha_{ij}^{(l)}$ is the aggregation weight for the message passing from node $v_j$ to node $v_i$. Existing GCNs mainly differ in the mechanism for computing $\alpha_{ij}^{(l)}$ (Kipf & Welling, 2017; Veličković et al., 2018; Ye et al., 2020; Hamilton et al., 2017; Zhang et al., 2020).

**Stochastic Block Model (SBM).** SBM (Holland et al., 1983) is a generative model for producing graphs with community structures. It parameterizes the edge probability between each node pair by

$$\bar{a}_{ij} | \mathbf{y}_i, \mathbf{y}_j \sim \begin{cases} \text{Bernoulli}(p_0), & \text{if } \mathbf{y}_i = \mathbf{y}_j \\ \text{Bernoulli}(p_1), & \text{if } \mathbf{y}_i \neq \mathbf{y}_j \end{cases}, \tag{2}$$

where $\bar{a}_{ij}$ is an indicator variable for the edge linking nodes $v_i$ and $v_j$, $\mathbf{y}_i$ and $\mathbf{y}_j$ denote their corresponding communities (classes), $p_0$ and $p_1$ are termed community link strength and cross-

community link probability, respectively. The case where $p_0 > p_1$ is called an assortative model, while the case $p_0 < p_1$ is called disassortative. In this paper, we leverage an assortative-constrained SBM (Gribel et al., 2020) with $p_0 = 1$ and $p_1 = 0$ to model the latent graph for a clear topology.

**Over-smoothing.** Real-world graphs often possess high sparsity and are corrupted by certain noise that leads to inter-class misconnection and missing intra-class edges. Over-smoothing is mainly caused by the indistinguishable features of nodes in different classes produced by the message passing along inter-class edges. Various strategies have been developed to alleviate this problem. JK-Net (Xu et al., 2018) utilizes skip connection for adaptive feature aggregation and DNA (Fey, 2019) further makes improvements based on the attention mechanism. PPNP and APPNP (Klicpera et al., 2019a) modify the message passing scheme by personalized PageRank (PPR) to avoid reaching the limit distribution of random walk. CGNN (Xhonneux et al., 2020) addresses over-smoothing in a similar manner as PPR. Zhao & Akoglu (2020) introduced a graph layer normalization scheme termed PairNorm to maintain the total pairwise distance between nodes unchanged across layers. GCNII (Chen et al., 2020b) extends GCN with Initial residual and Identity mapping. However, these methods cannot fundamentally address the over-smoothing issue, as they all view the observed graph as ground truth and the features of nodes in different classes would still be over-mixed along the inter-class edges. AdaEdge (Chen et al., 2020a) constantly refines the graph topology by adjusting the edges in a self-training-like fashion. However, AdaEdge only adjusts the edges linking nodes classified with high confidence, which leads to limited improvement or degradation in classification performance due to the incorrect operations for misclassified nodes. DropEdge (Rong et al., 2020) randomly removes a certain fraction of edges to reduce message passing. Despite enhanced robustness, DropEdge does not essentially optimize the graph topology. BBGDC (Hasanzadeh et al., 2020) generalizes Dropout (Srivastava et al., 2014) and DropEdge by adaptive connection sampling.

**Uncertain Graphs and Topology Optimization.** Learning with uncertain graphs is another related research area, where the observed graph structure is supposed to be derived from noisy data rather than ground truth. Bayesian approaches are typical methods that introduce uncertainty to network analysis. Zhang et al. (2019) developed BGCN that considers the observed graph as a sample from a parametric family of random graphs and makes maximum a posteriori (MAP) estimate of the graph parameters. Tiao et al. (2019) also viewed graph edges as Bernoulli random variables and used variational inference to optimize the posterior distribution of the adjacency matrix by approximating the pre-defined graph priors. Some other Bayesian methods have also been developed to combine GCNs with probabilistic models (Ng et al., 2018; Ma et al., 2019). However, without explicit optimization for the graph structure, they only improve the robustness under certain conditions such as incomplete edges, active learning, and adversarial attacks. For explicit topology optimization, Franceschi et al. (2019) presented LDS to parameterize edges as independent Bernoulli random variables and learn discrete structures for GCNs by solving a bilevel programming. However, LDS requires an extra validation set for training and suffers from limited scalability. TO-GCN (Yang et al., 2019) only adds the intra-class edges derived from the labeled nodes, which causes topology imbalance between $\mathcal{V}_u$ and $\mathcal{V}_l$. GDC (Klicpera et al., 2019b) refines the adjacency matrix with graph diffusion to consider the links between high-order neighborhoods. However, the added edges might still be noisy to hamper the classification. GRCN (Yu et al., 2020) modifies the original adjacency matrix by adding a residual matrix with each element measuring the similarity between two corresponding node embeddings, and IDGL (Chen et al., 2020c) iteratively learns the graph structure in a similar manner. Pro-GNN (Jin et al., 2020) introduces low rank and sparsity constraints to recover a clean graph in defending adversarial attacks. NeuralSparse (Zheng et al., 2020) uses the Gumbel Softmax trick (Jang et al., 2017) to sample $k$ neighbors from the original neighborhoods for each node but does not consider recovering missing intra-class edges. Different from the aforementioned methods, VEM-GCN aims at relieving the over-smoothing issue. We introduce a learned latent graph based on the assortative-constrained SBM to explicitly enhance intra-class connection and suppress inter-class interaction with the variational EM algorithm.

## 3 METHODOLOGY

In this section, we develop the VEM-GCN architecture for transductive node classification. VEM-GCN leverages the variational EM algorithm to achieve topology optimization, and consequently, address the over-smoothing issue by reducing noisy interactions between nodes in different classes. Specifically, E-step approximates the posterior probability distribution of the latent adjacency ma-

trix to optimize the graph structure, and M-step maximizes the evidence lower bound of the log-likelihood function based on the refined graph. We first introduce our motivation and provide an overview of the proposed VEM-GCN architecture. Subsequently, we elaborate the mechanisms of the variational E-step and M-step, respectively.

## 3.1 MOTIVATION AND OVERVIEW

**Motivation.** As mentioned above, a graph with its nodes densely connected within their own communities (classes) has lower risk of over-smoothing. Under this consideration, the optimal adjacency matrix for GCN is $\tilde{\mathbf{A}} = \mathbf{Y}\mathbf{Y}^\top$ (Yang et al., 2019; Chen et al., 2020a), where $\mathbf{Y} \in \mathbb{R}^{N \times C}$ is the matrix of one-hot-encoded ground-truth labels. However, since we have to infer $\mathbf{Y}_u$ for the unlabeled nodes $\mathcal{V}_u$, their true labels are not available for calculating $\tilde{\mathbf{A}}$. Thus, we introduce a latent graph $\mathbf{A}_{\text{latent}}$ learned from $\mathcal{G}_{\text{obs}}$ through another neural network to help generate a topology clearer than $\mathbf{A}_{\text{obs}}$ for GCNs. It is obvious that $\tilde{\mathbf{A}}$ is equivalent to a SBM with $p_0 = 1$ and $p_1 = 0$, and therefore we base the posterior probability distribution of the latent graph on this assumption.

**Overview.** The basic principle behind our proposed VEM-GCN architecture is maximum likelihood estimation (MLE) in a latent variable model, i.e., to maximize the log-likelihood function of the observed node labels $\mathbb{E}_{q_\phi(\mathbf{A}_{\text{latent}}|\mathcal{G}_{\text{obs}})}[\log p_\theta(\mathbf{Y}_l|\mathcal{G}_{\text{obs}})]$ based on the approximate posterior distribution $q_\phi(\mathbf{A}_{\text{latent}}|\mathcal{G}_{\text{obs}})$ of the latent graph $\mathbf{A}_{\text{latent}}$. According to variational inference, the evidence lower bound (ELBO) is optimized instead:

$$
\begin{aligned}
\log p_\theta(\mathbf{Y}_l|\mathcal{G}_{\text{obs}}) &\geq \mathcal{L}_{\text{ELBO}}(\theta, \phi; \mathbf{Y}_l, \mathcal{G}_{\text{obs}}) \\
&= \mathbb{E}_{q_\phi(\mathbf{A}_{\text{latent}}|\mathcal{G}_{\text{obs}})}[\log p_\theta(\mathbf{Y}_l, \mathbf{A}_{\text{latent}}|\mathcal{G}_{\text{obs}}) - \log q_\phi(\mathbf{A}_{\text{latent}}|\mathcal{G}_{\text{obs}})],
\end{aligned}
\tag{3}
$$

where the equality holds when $q_\phi(\mathbf{A}_{\text{latent}}|\mathcal{G}_{\text{obs}}) = p_\theta(\mathbf{A}_{\text{latent}}|\mathbf{Y}_l, \mathcal{G}_{\text{obs}})$. Note that $q_\phi$ can be arbitrary desirable distributions on $\mathbf{A}_{\text{latent}}$ and we use a neural network to parameterize it in this work. To jointly optimize the latent graph topology $\mathbf{A}_{\text{latent}}$ and the ELBO $\mathcal{L}_{\text{ELBO}}(\theta, \phi; \mathbf{Y}_l, \mathcal{G}_{\text{obs}})$, we adopt the variational EM algorithm to solve it (refer to Appendix A for the full algorithm).

## 3.2 E-STEP

In the inference procedure (E-step), $\theta$ is fixed and the goal is to optimze $q_\phi(\mathbf{A}_{\text{latent}}|\mathcal{G}_{\text{obs}})$ to approximate the true posterior distribution $p_\theta(\mathbf{A}_{\text{latent}}|\mathbf{Y}_l, \mathcal{G}_{\text{obs}})$. Under the condition of SBM, we assume each edge of the latent graph to be independent. Thus, $q_\phi(\mathbf{A}_{\text{latent}}|\mathcal{G}_{\text{obs}})$ can be factorized by:

$$
q_\phi(\mathbf{A}_{\text{latent}}|\mathcal{G}_{\text{obs}}) = \prod_{i,j} q_\phi(a_{ij}^{\text{latent}}|\mathcal{G}_{\text{obs}}).
\tag{4}
$$

Unlike LDS (Franceschi et al., 2019) using $\mathcal{O}(N^2)$ Bernoulli random variables to characterize the optimized graph with $N$ nodes, we parameterize $q_\phi(a_{ij}^{\text{latent}}|\mathcal{G}_{\text{obs}})$ through a neural network shared by all the possible node pairs (i.e., amortized variational inference (Gershman & Goodman, 2014)), as shown in Eq. 5. Hence, our method shows scalability for large-scale graphs and is easier to train.

$$
\mathbf{z}_i = \mathbf{NN}(\mathbf{e}_i), \quad q_\phi(a_{ij}^{\text{latent}} = 1|\mathcal{G}_{\text{obs}}) = \text{sigmoid}(\mathbf{z}_i \mathbf{z}_j^\top),
\tag{5}
$$

where $\mathbf{e}_i$ is the node embedding of node $v_i$, which can be derived from any desirable network embedding methods (e.g., node2vec (Grover & Leskovec, 2016), struc2vec (Ribeiro et al., 2017), and GCNs) or the raw node attributes $\mathbf{x}_i$ (the $i$-th row of $\mathbf{X}$), and $\mathbf{z}_i$ is the transformed features of node $v_i$. $\mathbf{NN}(\cdot)$ denotes a neural network and we use a Multi-Layer Perceptron (MLP) in this work. The probability for linking a node pair is defined as the inner-product of their transformed features activated by a sigmoid function.

To approximate the posterior probability distribution of $\mathbf{A}_{\text{latent}}$, we rewrite $p_\theta(\mathbf{A}_{\text{latent}}|\mathbf{Y}_l, \mathcal{G}_{\text{obs}})$ as:

$$
\begin{aligned}
p_\theta(\mathbf{A}_{\text{latent}}|\mathbf{Y}_l, \mathcal{G}_{\text{obs}}) &= \sum_{\mathbf{Y}_u} p_\theta(\mathbf{A}_{\text{latent}}, \mathbf{Y}_u|\mathbf{Y}_l, \mathcal{G}_{\text{obs}}) \\
&= \mathbb{E}_{p_\theta(\mathbf{Y}_u|\mathbf{Y}_l, \mathcal{G}_{\text{obs}})}[p_\theta(\mathbf{A}_{\text{latent}}|\mathbf{Y}_l, \mathbf{Y}_u, \mathcal{G}_{\text{obs}})].
\end{aligned}
\tag{6}
$$

Here, $p_\theta(\mathbf{A}_{\text{latent}}|\mathbf{Y}_l, \mathbf{Y}_u, \mathcal{G}_{\text{obs}})$ is parameterized by the aforementioned assortative-constrained SBM (i.e., $p_\theta(a_{ij}^{\text{latent}} = 1|\mathbf{y}_i, \mathbf{y}_j) = \mathbf{y}_i \mathbf{y}_j^\top$ for the one-hot-encoded node label $\mathbf{y}$), $p_\theta(\mathbf{Y}_u|\mathbf{Y}_l, \mathcal{G}_{\text{obs}})$ is the

predicted categorical distributions for the unlabeled nodes derived in the previous M-step. Consequently, we can sample $\hat{\mathbf{Y}}_u \sim p_\theta(\mathbf{Y}_u|\mathbf{Y}_l, \mathcal{G}_{\text{obs}})$ to estimate the expectation in Eq. 6 and leverage stochastic gradient descent (SGD) to minimize the reverse KL-divergence between the approximate posterior distribution $q_\phi(\mathbf{A}_{\text{latent}}|\mathcal{G}_{\text{obs}})$ and the target $p_\theta(\mathbf{A}_{\text{latent}}|\mathbf{Y}_l, \mathcal{G}_{\text{obs}})$. Under appropriate assumptions, $q_\phi$ will converge to $p_\theta(\mathbf{A}_{\text{latent}}|\mathbf{Y}_l, \mathcal{G}_{\text{obs}})$ as the iteration step of SGD $t \to \infty$ (Bottou, 2010). Thus, we can obtain the following objective function in the variational E-step for optimizing $\phi$:

$$\mathcal{L}_E = -\sum_{i,j} \sum_{a_{ij}^{\text{latent}} \in \{0,1\}} \lambda(a_{ij}^{\text{latent}}) p_\theta(a_{ij}^{\text{latent}}|\mathbf{y}_i, \mathbf{y}_j) \log q_\phi(a_{ij}^{\text{latent}}|\mathcal{G}_{\text{obs}}), \qquad (7)$$

where $\mathbf{y}$ is the ground truth label for node in labeled set $\mathcal{V}_l$, otherwise sampled from $p_\theta(\mathbf{Y}_u|\mathbf{Y}_l, \mathcal{G}_{\text{obs}})$ for the nodes without given labels in each training step, and $\lambda(a_{ij}^{\text{latent}})$ is the weighting hyperparameter to alleviate class imbalance between the inter-class edges and the intra-class edges.

### 3.3 M-step

In the learning procedure (M-step), $\phi$ is fixed and $\theta$ is updated to maximize the ELBO in Eq. 3. By factorizing $p_\theta(\mathbf{Y}_l, \mathbf{A}_{\text{latent}}|\mathcal{G}_{\text{obs}}) = p_{\theta_1}(\mathbf{Y}_l|\mathbf{A}_{\text{latent}}, \mathcal{G}_{\text{obs}}) p_{\theta_2}(\mathbf{A}_{\text{latent}}|\mathcal{G}_{\text{obs}})$ with $\theta = \{\theta_1, \theta_2\}$, we have:

$$\mathcal{L}_{\text{ELBO}} = \mathbb{E}_{q_\phi(\mathbf{A}_{\text{latent}}|\mathcal{G}_{\text{obs}})}[\log p_{\theta_1}(\mathbf{Y}_l|\mathbf{A}_{\text{latent}}, \mathcal{G}_{\text{obs}})] - \text{KL}[q_\phi(\mathbf{A}_{\text{latent}}|\mathcal{G}_{\text{obs}})\|p_{\theta_2}(\mathbf{A}_{\text{latent}}|\mathcal{G}_{\text{obs}})]. \quad (8)$$

Here, $p_{\theta_1}(\mathbf{Y}_l|\mathbf{A}_{\text{latent}}, \mathcal{G}_{\text{obs}})$ in the first term can be parameterized by arbitrary GCN models described by Eq. 1 that infer the node labels from $\mathbf{A}_{\text{latent}}$ and $\mathbf{X}$. We use the vanilla GCN (Kipf & Welling, 2017) in this work (see Eq. 13 in Appendix A). The second term is the KL-divergence between $q_\phi(\mathbf{A}_{\text{latent}}|\mathcal{G}_{\text{obs}})$ and the prior $p_{\theta_2}(\mathbf{A}_{\text{latent}}|\mathcal{G}_{\text{obs}})$, which can be optimized by setting $\theta_2 = \phi$ to force $\text{KL}[q_\phi(\mathbf{A}_{\text{latent}}|\mathcal{G}_{\text{obs}})\|p_{\theta_2}(\mathbf{A}_{\text{latent}}|\mathcal{G}_{\text{obs}})] = 0$. Actually, $p_{\theta_2}(\mathbf{A}_{\text{latent}}|\mathcal{G}_{\text{obs}})$ is of little interest to the final node classification task and we just need to maximize $\mathbb{E}_{q_\phi(\mathbf{A}_{\text{latent}}|\mathcal{G}_{\text{obs}})}[\log p_{\theta_1}(\mathbf{Y}_l|\mathbf{A}_{\text{latent}}, \mathcal{G}_{\text{obs}})]$ in the M-step.

Considering the fact that the observed graph structure $\mathbf{A}_{\text{obs}}$ should not be fully discarded and the approximation $q_\phi(\mathbf{A}_{\text{latent}}|\mathcal{G}_{\text{obs}})$ derived in the previous E-step is sometimes not very accurate, we use $q_\phi(\mathbf{A}_{\text{latent}}|\mathcal{G}_{\text{obs}})$ to refine $\mathbf{A}_{\text{obs}}$, and substitute $q_\phi$ with the following $\bar{q}_\phi$ in practice:

$$\bar{q}_\phi(a_{ij}^{\text{latent}} = 1|\mathcal{G}_{\text{obs}}) = \begin{cases} p, & \text{if } q_\phi > \varepsilon_1 \\ 0, & \text{if } q_\phi < \varepsilon_2 \\ p \cdot a_{ij}^{\text{obs}}, & \text{otherwise} \end{cases}, \qquad (9)$$

where $p \in (0, 1]$, $\varepsilon_1$ is close to one (commonly 0.999), and $\varepsilon_2$ is close to zero (commonly 0.01). Eq. 9 implies that, for edges predicted by $q_\phi$ to be linked with high confidence (the value after sigmoid or the maximum value after softmax), they should be added to the observed graph with probability $p$. Edges predicted by $q_\phi$ to be cut off with high confidence should be removed from the observed graph. Otherwise, we maintain the original graph structure with probability $p$.

Similar to the E-step, we can sample the latent adjacency matrix $\hat{\mathbf{A}}_{\text{latent}} \sim \bar{q}_\phi(\mathbf{A}_{\text{latent}}|\mathcal{G}_{\text{obs}})$ (note that we pre-train $p_{\theta_1}$ using $\mathbf{A}_{\text{obs}}$ and leverage SGD to minimize the cross-entropy error between the GCN's predictions $p_{\theta_1}(\mathbf{Y}_l|\hat{\mathbf{A}}_{\text{latent}}, \mathcal{G}_{\text{obs}})$ and the ground-truth labels $\mathbf{Y}_l$ for optimizing $\theta$:

$$\mathcal{L}_M = -\sum_{v_i \in \mathcal{V}_l} \sum_{c=1}^{C} y_{ic} \log p_{\theta_1}(y_{ic}|\hat{\mathbf{A}}_{\text{latent}}, \mathcal{G}_{\text{obs}}). \qquad (10)$$

In the test procedure, the final predictions for $\mathbf{Y}_u$ are $\mathbb{E}_{\bar{q}_\phi(\mathbf{A}_{\text{latent}}|\mathcal{G}_{\text{obs}})}[p_{\theta_1}(\mathbf{Y}_u|\mathbf{A}_{\text{latent}}, \mathcal{G}_{\text{obs}})]$, which can be approximated by Monte-Carlo sampling:

$$p_\theta(\mathbf{Y}_u|\mathbf{Y}_l, \mathcal{G}_{\text{obs}}) = \frac{1}{S} \sum_{i=1}^{S} p_{\theta_1}(\mathbf{Y}_u|\mathbf{A}_{\text{latent}}^i, \mathcal{G}_{\text{obs}}), \quad \text{with } \mathbf{A}_{\text{latent}}^i \sim \bar{q}_\phi(\mathbf{A}_{\text{latent}}|\mathcal{G}_{\text{obs}}), \qquad (11)$$

where the number of samples $S$ and the probability $p$ in $\bar{q}_\phi$ are tuned hyperparameters.

The two neural networks $q_\phi$ and $p_\theta$ are trained in an alternating fashion to reinforce each other. Topology optimization in the E-step improves the performance of the GCN in the M-step, and with more unlabeled nodes being correctly classified, $q_\phi$ will better approximate the optimal graph $\tilde{\mathbf{A}}$.

### 3.4 DISCUSSIONS

In this subsection, we discuss the relationship between VEM-GCN and two recent works for tackling over-smoothing, i.e., DropEdge (Rong et al., 2020) and AdaEdge (Chen et al., 2020a). We show that these two methods are specific cases of VEM-GCN under certain conditions. More detailed comparisons with other related works (e.g., SBM-related GCNs) are discussed in Appendix B.

**VEM-GCN vs. DropEdge.** DropEdge randomly removes a certain fraction of edges in each training step. The authors proved that this strategy can retard the convergence speed of over-smoothing. However, it does not address the over-smoothing issue at the core, since the graph topology is not fundamentally optimized and noisy messages still pass along inter-class edges. Considering the scenario where a node has few interactions with its community but many cross-community links, DropEdge cannot improve the discrimination of this stray node, since it does not recover the missing intra-class edges. We find that VEM-GCN degenerates to DropEdge, if we skip the E-step and just maximize $\mathbb{E}_{\bar{q}_\phi(\mathbf{A}_{\text{latent}}|\mathcal{G}_{\text{obs}})}[\log p_{\theta_1}(\mathbf{Y}_l|\mathbf{A}_{\text{latent}}, \mathcal{G}_{\text{obs}})]$ with $\bar{q}_\phi(a_{ij}^{\text{latent}} = 1|\mathcal{G}_{\text{obs}}) = p \cdot a_{ij}^{\text{obs}}$.

**VEM-GCN vs. AdaEdge.** AdaEdge also constantly adjusts the graph topology in the training procedure. It adds the edge between two nodes which are predicted by the GCN as the same class with high confidence, and removes edges in a similar manner. If we skip the E-step and set $\bar{q}_\phi$ as Eq. 12, VEM-GCN and AdaEdge can be equivalent.

$$\bar{q}_\phi(a_{ij}^{\text{latent}} = 1|\mathcal{G}_{\text{obs}}) = \begin{cases} 1, & \text{if } \mathbf{y}_i' = \mathbf{y}_j' \text{ and } \text{conf}(\mathbf{y}_i'), \text{conf}(\mathbf{y}_j') > \tau_1 \\ 0, & \text{if } \mathbf{y}_i' \neq \mathbf{y}_j' \text{ and } \text{conf}(\mathbf{y}_i'), \text{conf}(\mathbf{y}_j') > \tau_2 \\ a_{ij}^{\text{obs}}, & \text{otherwise} \end{cases}, \quad (12)$$

where $\mathbf{y}'$ is the prediction made by GCN, $\text{conf}(\cdot)$ denotes the corresponding confidence, $\tau_1$ and $\tau_2$ are two thresholds. Eq. 12 implies that, this self-training-like fashion only adjusts the edges whose interacting nodes have already been classified with high confidence. Therefore, the performance improvement is limited and would even get worse for some misclassified nodes, as it might wrongly add inter-class edges to the observed graph $\mathbf{A}_{\text{obs}}$ and remove helpful intra-class connections.

## 4 EXPERIMENTS

To evaluate our VEM-GCN architecture, we conduct extensive experiments on seven benchmark datasets. Under the same setting as DropEdge (Rong et al., 2020) and a label-scarce setting (i.e., low label rate), we compare the performance of VEM-GCN against a variety of state of the arts for tackling over-smoothing, uncertain graphs and topology optimization in GCNs. We further give the visualization results of topology optimization and quantitative analysis to verify the effectiveness of VEM-GCN in relieving the over-smoothing issue (complexity analysis is provided in Appendix E.3).

### 4.1 EXPERIMENTAL SETUP

**Datasets and Baselines.** We adopt seven well-known benchmark datasets to validate the proposed method. Cora (Sen et al., 2008), Cora-ML (McCallum et al., 2000; Bojchevski & Günnemann, 2018), Citeseer (Sen et al., 2008), and Pubmed (Namata et al., 2012) are four citation network benchmarks, where nodes represent documents and edges are citations between documents. Amazon Photo and Amazon Computers are two segments from the Amazon co-purchase graph (McAuley et al., 2015), in which nodes represent goods and edges indicate that two goods are frequently bought together. In the Microsoft Academic graph (Shchur et al., 2018), nodes are authors and edges represent their co-authorship. All graphs use bag-of-words encoded representations as node attributes. An overview of the dataset statistics is summarized in Appendix C.

Since VEM-GCN aims at addressing the over-smoothing problem with topology optimization, we evaluate the node classification performance of our method against various strategies for tackling over-smoothing, uncertain graphs and topology optimization in GCNs. For addressing the over-smoothing issue, five methods are considered: DropEdge (Rong et al., 2020), DropICE, AdaEdge (Chen et al., 2020a), PairNorm (Zhao & Akoglu, 2020), and BBGDC (Hasanzadeh et al., 2020), in which DropICE is implemented by removing the inter-class edges derived from $\mathcal{V}_l$. For tackling uncertain graphs, we compare against several Bayesian approaches including BGCN (Zhang et al., 2019), VGCN (Tiao et al., 2019), and G$^3$NN (Ma et al., 2019). For topology optimization, LDS

Table 1: Average test accuracy (%) for all models (a two-layer vanilla GCN as the backbone) and all datasets under the full-supervised setting. OOM: Out-of-memory error.

| Method | Cora | Citeseer | Pubmed | Cora-ML | Amazon Photo | Amazon Computers | MS Academic |
|---|---|---|---|---|---|---|---|
| Vanilla GCN | 87.5 | 78.6 | 88.4 | 90.6 | 93.6 | 88.5 | 93.5 |
| AdaEdge | 87.6 | 78.9 | 88.2 | 90.8 | 93.8 | 88.4 | 93.7 |
| DropEdge | 87.8 | 79.0 | 88.4 | 90.7 | 93.9 | 88.6 | 93.8 |
| DropICE | 86.8 | 78.3 | 87.2 | 90.5 | 93.5 | 86.3 | 92.5 |
| PairNorm | 85.2 | 77.2 | 87.6 | 88.8 | 94.4 | 89.9 | 91.7 |
| BBGDC | 87.0 | 77.2 | OOM | 90.4 | 91.2 | OOM | OOM |
| LDS | 87.4 | 79.6 | OOM | 91.0 | 94.1 | OOM | OOM |
| TO-GCN | 86.3 | 77.5 | 88.1 | 89.0 | 92.7 | 87.3 | 92.3 |
| GDC | 86.2 | 77.1 | 87.2 | 90.2 | 93.9 | **92.2** | 93.8 |
| GRCN | 87.8 | 76.7 | 86.9 | 90.2 | 91.6 | 77.1 | 92.8 |
| IDGL | 88.2 | 79.0 | 88.1 | 91.3 | 94.2 | 87.9 | OOM |
| BGCN | 87.9 | 78.7 | 88.0 | 91.1 | 93.1 | 87.8 | 92.8 |
| VGCN | 87.3 | 78.4 | 87.8 | 90.5 | 93.5 | 88.1 | 93.3 |
| G$^3$NN | 87.5 | 77.6 | 88.7 | 90.7 | 94.8 | 90.1 | 94.0 |
| GMNN | 87.2 | 78.7 | 88.3 | 90.4 | 93.8 | 90.5 | 93.5 |
| **VEM-GCN** | **88.7** | **80.6** | **90.1** | **91.6** | **95.7** | 91.8 | **95.5** |

(Franceschi et al., 2019), GDC (Klicpera et al., 2019b), TO-GCN (Yang et al., 2019), GRCN (Yu et al., 2020), and IDGL (Chen et al., 2020c) are the baselines. GMNN (Qu et al., 2019) is also taken as a baseline, as it also employs variational EM for transductive node classification.

We conduct node classification under two experimental settings, i.e., full-supervised and label-scarce settings. The full-supervised setting follows DropEdge (Rong et al., 2020), where each dataset is split into 500 nodes for validation, 1000 nodes for test and the rest for training. The label-scarce setting assigns labels to only a few nodes and selects 500 nodes for validation, while the rest are used for test. Under the label-scarce setting, we compare VEM-GCN with the baselines except for LDS, as LDS always uses the validation set for training, which is unfair for learning with limited training samples. DropICE is also omitted since the number of the removed inter-class edges derived from $\mathcal{V}_l$ is very small in the label-scarce setting and thus DropICE only obtains similar performance as the vanilla GCN. Considering that the classification performance is highly influenced by the split of the dataset (Shchur et al., 2018), we run all the models with the same 5 random data splits for each evaluation. To further ensure the credibility of the results, we perform 10 random weight initializations for each data split and report the average test accuracy for both experimental settings.

**Model Configurations.** For a fair comparison, we evaluate all the methods under the same GCN backbone and the same training procedure. To be concrete, the graph model used in all baselines and our VEM-GCN ($p_{\theta_1}$ in the M-step) is a vanilla GCN (Kipf & Welling, 2017) with the number of hidden units set as 32. Besides, we train the GCN backbone of all the methods for each dataset with the same dropout rate of 0.5, the same weight decay, the same learning rate of 0.01, the same optimizer (Adam (Kingma & Ba, 2015)), the same maximum training epoch of 1500, and the same early stopping strategy based on the validation loss with a patience of 50 epochs (for deeper models with more than 2 layers, we set the patience as 100 epochs). Note that IDGL empirically needs more training epochs to converge and we set its maximum training epoch as 10000 with a patience of 1000 epochs. As for $q_\phi$ in the E-step, the input node embeddings are the attributes averaged over the neighborhood of each node and the network architecture is a four-layer MLP with hidden units of size 128, 64, 64, and 32, respectively. Please refer to Appendix D for more details of the implementations and hyperparameter settings for each dataset.

## 4.2 RESULTS AND ANALYSIS

**Full-supervised Setting.** Table 1 summarizes the classification results. The highest accuracy in each column is highlighted in bold. Note that the results of BBGDC and LDS on three large graphs (i.e., Pubmed, Amazon Computers, and MS Academic) and IDGL on MS Academic graph are missing due to the out-of-memory error. Table 1 demonstrates that none of the baselines outperform the vanilla GCN in all cases, while VEM-GCN consistently improves the test accuracy of the GCN

Table 2: Average test accuracy (%) and over-smoothness under the label-scarce setting (10 labels per class) with varying layers. For both metrics, the larger the better. A: Accuracy. S: Over-smoothness.

| Dataset | Method | 2 layers | 4 layers | 6 layers | 8 layers | 10 layers |
|---|---|---|---|---|---|---|
| Cora | Vanilla GCN [A / S] | 74.5 / 2.39 | 73.2 / 2.09 | 68.8 / 2.04 | 66.7 / 2.01 | 46.9 / 1.73 |
| | DropEdge [A / S] | 75.0 / 2.01 | 75.5 / 2.59 | 58.1 / 2.10 | 42.7 / 1.84 | 32.1 / 1.46 |
| | AdaEdge [A / S] | 74.9 / 2.47 | 73.9 / 2.42 | 72.0 / 2.47 | 68.9 / 2.05 | 54.6 / 1.86 |
| | **VEM-GCN** [A / S] | **77.7 / 2.51** | **78.0 / 3.57** | **78.4 / 2.51** | **78.1 / 3.11** | **78.1 / 2.94** |
| Citeseer | Vanilla GCN [A / S] | 61.0 / 1.82 | 56.7 / 1.83 | 53.7 / 1.75 | 44.9 / 1.66 | 26.7 / 1.51 |
| | DropEdge [A / S] | 60.3 / 1.83 | 56.5 / 1.95 | 50.1 / 1.52 | 35.5 / 1.24 | 23.7 / 1.13 |
| | AdaEdge [A / S] | 61.5 / 1.84 | 58.7 / 1.83 | 53.1 / 1.87 | 49.1 / 1.72 | 43.5 / 1.62 |
| | **VEM-GCN** [A / S] | **64.2 / 1.89** | **64.2 / 2.01** | **63.7 / 1.85** | **63.8 / 1.93** | **63.7 / 1.83** |
| Cora-ML | Vanilla GCN [A / S] | 83.4 / 2.87 | 81.3 / 3.15 | 77.7 / 2.46 | 66.4 / 2.11 | 44.9 / 1.73 |
| | DropEdge [A / S] | 83.1 / 2.85 | 81.4 / 3.17 | 77.6 / 2.42 | 43.4 / 2.59 | 37.1 / 2.39 |
| | AdaEdge [A / S] | 83.3 / 2.98 | 81.0 / 3.21 | 78.1 / 2.89 | 70.2 / 2.50 | 53.5 / 2.43 |
| | **VEM-GCN** [A / S] | **84.4 / 3.78** | **84.4 / 3.88** | **84.4 / 3.70** | **84.4 / 4.45** | **84.3 / 4.12** |

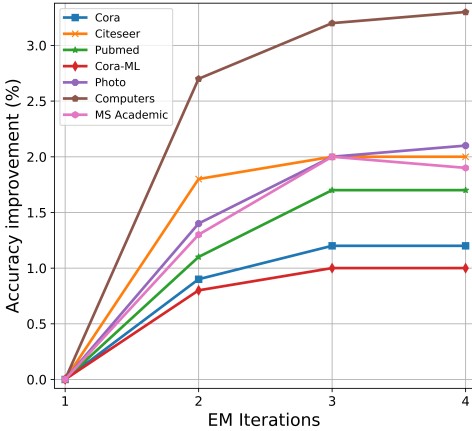

Figure 1: Convergence analysis.

Table 3: Average test accuracy (%) under different label rates on the Amazon Photo dataset.

| # labels per class | 5 | 10 | 20 | 30 |
|---|---|---|---|---|
| Vanilla GCN | 86.7 | 88.8 | 90.6 | 91.8 |
| AdaEdge | 86.5 | 88.9 | 90.5 | 91.8 |
| DropEdge | 86.5 | 88.8 | 90.5 | 91.6 |
| PairNorm | 78.6 | 83.7 | 86.2 | 88.1 |
| BBGDC | 87.3 | 88.4 | 90.1 | 90.4 |
| TO-GCN | 83.2 | 85.4 | 86.7 | 88.2 |
| GDC | 85.4 | 88.1 | 90.2 | 91.0 |
| GRCN | 84.1 | 88.3 | 90.4 | 91.9 |
| IDGL | 86.8 | 89.2 | 91.0 | 91.4 |
| BGCN | 85.1 | 87.1 | 89.1 | 91.1 |
| VGCN | 85.9 | 88.5 | 90.5 | 91.3 |
| G$^3$NN | 86.3 | 88.8 | 90.6 | 90.8 |
| GMNN | 87.9 | 89.4 | 91.2 | 92.3 |
| **VEM-GCN** | **89.2** | **90.5** | **91.8** | **92.8** |

backbone by noticeable margins. Specifically, we find the following facts under the full-supervised setting: (1) For tackling over-smoothing, AdaEdge, DropEdge and PairNorm demonstrate limited improvement on several datasets, while BBGDC and DropICE almost collapse for all cases. (2) LDS, TO-GCN, GDC, GRCN and IDGL cannot guarantee that their topology optimization could achieve performance gains for the GCN backbone. (3) Only adding intra-class edges (TO-GCN) or removing inter-class edges (DropICE) derived from $\mathcal{V}_l$ might cause topology imbalance between $\mathcal{V}_u$ and $\mathcal{V}_l$. The GCN trained on $\mathcal{V}_l$ with enhanced graph topology would fail on $\mathcal{V}_u$ with the original graph topology. (4) Bayesian approaches and GMNN can only achieve comparable performance with the vanilla GCN in almost all cases. Overall, these facts imply that VEM-GCN significantly benefits from the large labeled data to generate a clearer topology and achieve better performance.

**Label-scarce Setting.** We randomly select 10 labeled nodes per class as the training set and evaluate the performance of VEM-GCN with varying layers. Table 2 shows the test accuracy and the over-smoothness measurements of the learned node embeddings (input node features of the last layer). The metric to measure the over-smoothness is defined in Appendix D.3 and supplementary results of VEM-GCN on additional datasets are shown in Appendix E.1. As can be seen in Table 2, the vanilla GCN severely suffers from the over-smoothing issue, while VEM-GCN can achieve performance gains even with deeper layers (e.g., on the Cora dataset). DropEdge and AdaEdge can relieve the over-smoothing issue to some extent, but the performance still decreases drastically when stacking more GCN layers. The results of over-smoothness measurements indicate that VEM-GCN indeed produces more separable node embeddings across different classes to address the over-smoothing

Figure 2: Visualization results of topology optimization on the Cora-ML dataset under the full-supervised setting. We plot an induced subgraph (node indices from $450$ to $850$) for a better view. (a) The observed graph $\mathbf{A}_{\text{obs}}$; (b) The optimal graph $\tilde{\mathbf{A}}$; (c) The approximate posterior distribution $q_\phi$ on $\mathbf{A}_{\text{latent}}$; (d) The refined graph $\bar{q}_\phi$.

problem. We further take Amazon Photo as an example dataset to validate VEM-GCN under different label rates. Similar trend as Table 1 can be found in Table 3.

**Convergence Analysis and Visualization Results.** VEM-GCN leverages the variational EM algorithm for optimization. In this subsection, we analyze the convergence of VEM-GCN. Figure 1 depicts the accuracy improvement curve of $p_{\theta_1}$ during the EM iterations under the full-supervised setting. We find that VEM-GCN requires only a few iterations to converge. We further take Cora-ML as an example to give the corresponding visualization results of graph topology optimization. Figure 2 show that the observed graph is very sparse and contains a few intra-class edges, while the optimized graph recover many missing intra-class edges to relieve the over-smoothing problem. Note that, although the refined graph is much denser than the observed graph, the hyperparameter $p$ (0.05 here) in $\bar{q}_\phi$ helps maintain the sparsity of the latent adjacency matrix in the training procedure. Thus, the M-step can still be implemented efficiently using sparse-dense matrix multiplications.

## 5 CONCLUSION

In this paper, we present a novel architecture termed VEM-GCN for addressing the over-smoothing problem in GCNs with graph topology optimization. By introducing a latent graph parameterized by the assortative-constrained stochastic block model and utilizing the variational EM algorithm to jointly optimize the graph structure and the likelihood function, VEM-GCN outperforms a variety of state-of-the-art methods for tackling over-smoothing, uncertain graphs, and topology optimization in GCNs. For future work, we expect further improvements for the VEM-GCN architecture to deal with more complex graphs such as hypergraphs and heterogeneous graphs.

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

## A  ALGORITHM

For a fair comparison, we adopt the vanilla GCN (Kipf & Welling, 2017) as the backbone for all baselines and our proposed VEM-GCN architecture. A two-layer GCN is in the following form:

$$p_{\theta_1}(\mathbf{Y}|\mathbf{A}, \mathbf{X}) = \mathrm{softmax}\left(\dot{\mathbf{D}}^{-\frac{1}{2}}\dot{\mathbf{A}}\dot{\mathbf{D}}^{-\frac{1}{2}}\,\mathrm{ReLU}\left(\dot{\mathbf{D}}^{-\frac{1}{2}}\dot{\mathbf{A}}\dot{\mathbf{D}}^{-\frac{1}{2}}\mathbf{X}\mathbf{\Theta}^{(0)}\right)\mathbf{\Theta}^{(1)}\right), \qquad (13)$$

where $\mathbf{X}$ is the input node attribute matrix, $\mathbf{I}_N$ is the identity matrix, $\dot{\mathbf{A}} = \mathbf{A} + \mathbf{I}_N$ is the adjacency matrix with added self-loops, $\dot{\mathbf{D}}$ is its corresponding diagonal degree matrix, and $\theta_1 = \{\mathbf{\Theta}^{(0)}, \mathbf{\Theta}^{(1)}\}$ are the learnable weight parameters. Algorithm 1 describes the proposed VEM-GCN architecture.

## B  FURTHER DISCUSSIONS

In addition to recent strategies for tackling over-smoothing issues, we further distinguish VEM-GCN from the SBM-related GCNs and VGCN and GMNN (that introduce variational inference).

**Comparison with SBM-related GCNs.** Stochastic block model (SBM) has been combined with GCNs in several recent works (i.e., BGCN (Zhang et al., 2019) and G$^3$NN (Ma et al., 2019)). However, these architectures are totally different from VEM-GCN in motivations, objective functions and training methods. BGCN (Zhang et al., 2019) jointly infers the node labels and the parameters of SBM using only $\mathbf{A}_{\mathrm{obs}}$, which ignores the dependence of the graph on $\mathbf{X}$ and $\mathbf{Y}_l$. Then the

---

**Algorithm 1** VEM-GCN

---

**Input**: Observed graph $\mathcal{G}_{\text{obs}}$ and labels $\mathbf{Y}_l$ for the labeled nodes $\mathcal{V}_l$.
**Parameter**: $\phi$ in the E-step and $\theta$ in the M-step.
**Output**: Predicted labels $\mathbf{Y}_u$ for the unlabeled nodes $\mathcal{V}_u$.
1: Pre-train $p_\theta$ with $\mathbf{A}_{\text{obs}}$ and $\mathbf{Y}_l$ to get initial $p_\theta(\mathbf{Y}_u|\mathbf{Y}_l, \mathcal{G}_{\text{obs}})$.
2: **for** EM iteration $t = 1, \ldots, T$ **do**
3:    **E-step**:
4:    **for** training step $s_1 = 1, \ldots, S_1$ **do**
5:       Sample $\hat{\mathbf{Y}}_u \sim p_\theta(\mathbf{Y}_u|\mathbf{Y}_l, \mathcal{G}_{\text{obs}})$ for the unlabeled nodes $\mathcal{V}_u$.
6:       Set $p_\theta(\mathbf{A}_{\text{latent}}|\mathbf{Y}_l, \mathcal{G}_{\text{obs}}) = p_\theta(\mathbf{A}_{\text{latent}}|\mathbf{Y}_l, \hat{\mathbf{Y}}_u, \mathcal{G}_{\text{obs}})$ according to Eq. 6.
7:       Update $q_\phi$ to optimize the objective function in Eq. 7 with SGD.
8:    **end for**
9:    **M-step**:
10:   Obtain $\bar{q}_\phi(\mathbf{A}_{\text{latent}}|\mathcal{G}_{\text{obs}})$ according to Eq. 9.
11:   **for** training step $s_2 = 1, \ldots, S_2$ **do**
12:      Sample $\hat{\mathbf{A}}_{\text{latent}} \sim \bar{q}_\phi(\mathbf{A}_{\text{latent}}|\mathcal{G}_{\text{obs}})$ for the latent adjacency matrix.
13:      Update $p_\theta$ to maximize the log-likelihood $\log p_\theta(\mathbf{Y}_l|\hat{\mathbf{A}}_{\text{latent}}, \mathcal{G}_{\text{obs}})$ with SGD.
14:   **end for**
15:   Predict categorical distributions $p_\theta(\mathbf{Y}_u|\mathbf{Y}_l, \mathcal{G}_{\text{obs}})$ according to Eq. 11.
16: **end for**
17: **return** Final predicted labels for $\mathcal{V}_u$ based on $p_\theta(\mathbf{Y}_u|\mathbf{Y}_l, \mathcal{G}_{\text{obs}})$.

---

adjacency matrices sampled from the inferred SBM are used to train the GCN. Different from VEM-GCN, BGCN neither explicitly promotes intra-class connection nor demotes inter-class interaction. It only achieves robustness under certain conditions such as adversarial attacks, benefiting from the uncertainty brought by the inferred SBM. G$^3$NN is a flexible generative model, where the graph generated by SBM is based on the predictions of an additional MLP learned from only $\mathbf{X}$ and $\mathbf{Y}_l$. The predictions for the unlabeled nodes are still based on $\mathcal{G}_{\text{obs}}$ (i.e., the input adjacency matrix of the GCN is still $\mathbf{A}_{\text{obs}}$). By contrast, VEM-GCN aims at addressing the over-smoothing issue with topology optimization. In VEM-GCN, the M-step trains a GCN to obtain the predictions of the unlabeled nodes based on $\mathbf{A}_{\text{latent}}$, $\mathbf{X}$, and $\mathbf{Y}_l$. We then estimate the posterior distribution on $\mathbf{A}_{\text{latent}}$ based on $\mathbf{Y}_l$ and the predictions for the unlabeled nodes under the SBM assumption. Subsequently, the E-step optimizes the graph topology by training another auxiliary neural network with node embeddings as input to approximate the posterior distribution of $\mathbf{A}_{\text{latent}}$. The E-step and M-step are optimized in an alternating fashion to improve each other.

**VEM-GCN vs. VGCN.** VGCN (Tiao et al., 2019) also introduces a latent graph $\mathbf{A}_{\text{latent}}$ and optimizes $\mathcal{L}_{\text{ELBO}}$ in Eq. 3. However, it directly optimizes the ELBO in a VAE (Kingma & Welling, 2014) fashion and the posterior distribution of $\mathbf{A}_{\text{latent}}$ is set to approximate the pre-defined graph priors $p(a_{ij}^{\text{prior}} = 1) = \rho_1 a_{ij}^{\text{obs}} + \rho_2(1 - a_{ij}^{\text{obs}})$ with $0 < \rho_1, \rho_2 < 1$ using the re-parameterization trick. VGCN is to achieve robustness under fake link attacks and only shows comparable performance with GCN under the standard transductive learning setting (i.e., inferring $\mathbf{Y}_u$ based on the original $\mathcal{G}_{\text{obs}}$). By contrast, VEM-GCN does not introduce priors over graphs. We optimize the graph topology by explicitly enhancing intra-class connection and suppressing inter-class interaction using SBM and variational EM to relieve the over-smoothing issue.

**VEM-GCN vs. GMNN.** Graph Markov Neural Network (GMNN) (Qu et al., 2019) also employs variational EM for node classification, but it is totally different from our method in motivations and objective functions. GMNN focuses on modeling the joint distribution of object (node) labels. Therefore, GMNN views $\mathbf{Y}_u$ as latent variables and optimizes the following ELBO:

$$\log p_\theta(\mathbf{Y}_l|\mathbf{X}) \geq \mathbb{E}_{q_\phi(\mathbf{Y}_u|\mathbf{X})}[\log p_\theta(\mathbf{Y}_l, \mathbf{Y}_u|\mathbf{X}) - \log q_\phi(\mathbf{Y}_u|\mathbf{X})]. \tag{14}$$

In the E-step, GMNN parameterizes $q_\phi(\mathbf{Y}_u|\mathbf{X})$ with a GCN and $q_\phi(\mathbf{Y}_u|\mathbf{X})$ is optimized to approximate the posterior distribution $p_\theta(\mathbf{Y}_u|\mathbf{Y}_l, \mathbf{X})$. In the M-step, GMNN utilizes another GCN to model the conditional distribution $p_\theta(\mathbf{y}_i|\mathbf{y}_{\text{NB}(i)}, \mathbf{X})$ for each node $v_i \in \mathcal{V}$ (NB$(i)$ is the neighbor set of node $v_i$) with a conditional random field and maximizes the corresponding likelihood. On the contrary, VEM-GCN is proposed to relieve the over-smoothing issue. VEM-GCN optimizes the

Table 4: Summary of dataset statistics

| | Undirected graph | | | Directed graph | | | |
| --- | --- | --- | --- | --- | --- | --- | --- |
| **Dataset** | Cora | Citeseer | Pubmed | Cora-ML | Amazon Photo | Amazon Computers | MS Academic |
| **# Nodes** | 2708 | 3312 | 19717 | 2995 | 7650 | 13752 | 18333 |
| **# Edges** | 5278 | 4536 | 44324 | 8416 | 143662 | 287209 | 163788 |
| **# Features** | 1433 | 3703 | 500 | 2879 | 745 | 767 | 6805 |
| **# Classes** | 7 | 6 | 3 | 7 | 8 | 10 | 15 |

latent graph to approximate its posterior distribution based on SBM in the E-step, and trains a GCN based on the latent graph in the M-step.

## C  DATASET STATISTICS

We utilize seven node classification benchmarks in this paper, including four citation networks (i.e., Citeseer, Pubmed, Cora, and Cora-ML), two Amazon co-purchase graphs (i.e., Amazon Photo and Amazon Computers), and one Microsoft Academic graph, as summarized below.

- **Citation Networks**. Cora, Citeseer, Pubmed can be downloaded from the official source code of GCN (Kipf & Welling, 2017) publicly available at `https://github.com/tkipf/gcn/tree/master/gcn/data`, and Cora-ML can be downloaded from the source code of (A)PPNP (Klicpera et al., 2019a) publicly available at `https://github.com/klicperajo/ppnp/tree/master/ppnp/data`.

- **Amazon Co-purchase Graph**. The Amazon Photo and Amazon Computers datasets from the Amazon co-purchase graph can be publicly downloaded from `https://github.com/shchur/gnn-benchmark/tree/master/data` (Shchur et al., 2018).

- **Microsoft Academic Graph**. The MS Academic graph can be downloaded from the source code of (A)PPNP (Klicpera et al., 2019a) publicly available at `https://github.com/klicperajo/ppnp/tree/master/ppnp/data`.

An overview of the dataset statistics is listed in Table 4. Note that for these open datasets, three (Cora, Citeseer, Pubmed) are given in the form of undirected graphs, while four (Cora-ML, Amazon Photo, Amazon Computers, MS Academic) are directed graphs. GCN treats all these datasets as undirected graphs (i.e., $a'_{ij} = [a_{ij} + a_{ji} > 0]$, where $[\cdot]$ denotes Iverson bracket).

## D  FURTHER EXPERIMENTAL DETAILS

### D.1  IMPLEMENTATIONS

The implementation of VEM-GCN consists of two alternating steps in each iteration, including a variational E-step and an M-step. In the variational E-step, a simple four-layer MLP is implemented for $q_\phi$, where the numbers of neuron units of each layer are 128, 64, 64, and 32, respectively. We use *tanh* as the nonlinear activation function for the hidden layers. In the M-step, $p_{\theta_1}$ is a vanilla GCN with the number of hidden units set as 32, and we use the official source code from `https://github.com/tkipf/gcn/tree/master/gcn`. All the baselines and our VEM-GCN architecture are trained on a single NVIDIA GTX 1080 Ti GPU with 11GB memory.

We just utilize the raw node attributes $\mathbf{X}$ as the input to $q_\phi$. Note that we can also support any other desirable network embedding method and these experiments is left for the future work. Considering the fact that the bag-of-words representations of $\mathbf{X}$ is often noisy, we average the features of each node over its neighborhoods to smooth the input signal. Let $\hat{\mathbf{A}}_{\text{row}} = (\mathbf{D} + \gamma\mathbf{I}_N)^{-1}(\mathbf{A} + \gamma\mathbf{I}_N)$ denote the "self-enhanced" adjacency matrix with row normalization (we use $\gamma = 1.5$ in this paper), and $[\mathbf{A}\|\mathbf{B}]$ be the concatenation of matrices $\mathbf{A}$ and $\mathbf{B}$ along the last dimension. Consequently, we summarize the specific input to $q_\phi$ for all the datasets as below.

- For Cora, Citeseer, and MS Academic, we use $\mathbf{X}' = \hat{\mathbf{A}}_{\text{row}}\mathbf{X}$ as the input of $q_\phi$.

- For Pubmed and Amazon Photo, we use $\mathbf{X}' = \left[ \mathbf{X} \| \hat{\mathbf{A}}_{\text{row}} \mathbf{X} \| \hat{\mathbf{A}}_{\text{row}}^2 \mathbf{X} \right]$ as the input of $q_\phi$.

- For Cora-ML and Amazon Computers, we use $\mathbf{X}' = \left[ \mathbf{X} \| \hat{\mathbf{A}}_{\text{row}} \mathbf{X} \right]$ as the input of $q_\phi$.

For the sampling in the E-step, we find that it is not always stable to draw the sampled labels $\hat{\mathbf{Y}}_u \sim p_\theta \left( \mathbf{Y}_u | \mathbf{Y}_l, \mathcal{G}_{\text{obs}} \right)$. To alleviate this problem, we maintain $\hat{\mathbf{Y}}_u \leftarrow \operatorname{argmax}_y (p_\theta \left( \mathbf{Y}_u | \mathbf{Y}_l, \mathcal{G}_{\text{obs}} \right))$ with probability $p_e$, and sample $\hat{\mathbf{Y}}_u \sim p_\theta \left( \mathbf{Y}_u | \mathbf{Y}_l, \mathcal{G}_{\text{obs}} \right)$ with probability $(1 - p_e)$ to improve the performance in practice. For training of the E-step, we utilize SGD with momentum (of 0.99).

In the test procedure, we perform inference in two ways: (1) keep $p$ in Eq. 9 the same as the training procedure (commonly $p < 1$), sample $S$ ($S > 1$) adjacency matrices, and predict the classes for the unlabeled nodes according to Eq. 11; (2) set $p = 1$ (i.e., the latent adjacency matrix is now deterministic) and $S = 1$. We report the best test accuracy obtained using these two sampling methods on each dataset and find that (2) almost always performs better.

## D.2 Hyperparameter Settings

Table 5 summarizes the hyperparameters adopted for the full-supervised setting on the seven benchmark datasets. For the label-scarce setting, the hyperparameters are the same, except for $\varepsilon_1$ and $\varepsilon_2$ that need to be carefully tuned in the search space: $\varepsilon_1 \in \{0.95, 0.99, 0.995, 0.999, 0.9995, 0.9999\}$, $\varepsilon_2 \in \{0.001, 0.005, 0.01, 0.05, 0.1\}$.

Table 5: Hyperparameter setting for the results in Table 1. $\lambda(a_{ij}^{\text{obs}} = 0) = 1$ and $\lambda(a_{ij}^{\text{obs}} = 1) = \lambda$. lr denotes learning rate. An exponential decay schedule is adopted for lr (E) with decay rate $d_r$ and decay step $d_s$ tuned: $d_r \in \{0.96, 0.97, 0.98, 0.99\}$, $d_s \in \{2500, 3000, 3500, 4000\}$. Batch size of the M-step is set to 96 (i.e., we randomly select 96 nodes for each step of optimizing Eq. 7). $p$ is tuned in the search space: $p \in \{0.001, 0.002, \dots, 0.01, 0.02, \dots, 0.1, 0.15, \dots, 1\}$.

| Datasets | weight decay | lr (M) | lr (E) | $p_e$ | $\varepsilon_1$ | $\varepsilon_2$ | $\lambda$ | $p$ | $S$ |
|---|---|---|---|---|---|---|---|---|---|
| **Cora** | 0.0001 | 0.01 | 0.001 | 0.6 | 0.99 | 0.01 | 20 | 0.25 | 10 |
| **Citeseer** | 0.0001 | 0.01 | 0.002 | 0.6 | 0.95 | 0.1 | 20 | 0.25 | 10 |
| **Pubmed** | 0.0001 | 0.01 | 0.002 | 0.8 | 0.99 | 0.01 | 20 | 0.1 | 10 |
| **Cora-ML** | 0.0001 | 0.01 | 0.001 | 0.85 | 0.9999 | 0.01 | 20 | 0.05 | 10 |
| **Amazon Photo** | 0.0001 | 0.01 | 0.001 | 0.85 | 0.999 | 0.01 | 25 | 0.25 | 10 |
| **Amazon Computers** | – | 0.01 | 0.002 | 0.8 | 0.999 | 0.01 | 25 | 0.15 | 10 |
| **MS Academic** | – | 0.01 | 0.002 | 0.8 | 0.999 | 0.005 | 25 | 0.002 | 10 |

## D.3 Metric for Measuring Over-smoothness

To address the over-smoothing issue, one would prefer to reduce the intra-class distance to make node features in the same class similar, and increase the inter-class distance to produce distinguishable representations for nodes in different classes. Hence, we use the ratio of average inter-class distance to average intra-class distance (Euclidean distance of input node features in the last layer) to measure the over-smoothness. Formally, given the learned node embeddings $\mathbf{H} = \{\mathbf{h}_i\}_{i=1}^N$ (the input node features of the last layer), we first calculate the distance matrix $\mathbf{D} = [d_{ij}] \in \mathbb{R}^{N \times N}$ with each entry defined as

$$d_{ij} = \left\| \frac{\mathbf{h}_i}{\|\mathbf{h}_i\|_2} - \frac{\mathbf{h}_i}{\|\mathbf{h}_i\|_2} \right\|_2, \tag{15}$$

where $\| \cdot \|_2$ denotes Euclidean norm. Next, we define the inter-class mask matrix and intra-class mask matrix as follows:

$$\mathbf{M}^{\text{inter}} = -\tilde{\mathbf{A}} + \mathbf{1}^{N \times N}, \tag{16}$$

$$\mathbf{M}^{\text{intra}} = \tilde{\mathbf{A}} - \mathbf{I}_N, \tag{17}$$

where $\tilde{\mathbf{A}} = \mathbf{Y}\mathbf{Y}^\top$ is the optimal graph and $\mathbf{1}^{N \times N}$ is a matrix of size $N \times N$ with all entries set to 1. Then we can obtain the inter-class distance matrix $\mathbf{D}^{\text{inter}} = [d_{ij}^{inter}] \in \mathbb{R}^{N \times N}$ and the intra-class distance matrix $\mathbf{D}^{\text{intra}} = [d_{ij}^{intra}] \in \mathbb{R}^{N \times N}$ by element-wise multiplication $\mathbf{D}$ with the mask

matrices:

$$\mathbf{D}^{\text{inter}} = \mathbf{D} \circ \mathbf{M}^{\text{inter}}, \tag{18}$$

$$\mathbf{D}^{\text{intra}} = \mathbf{D} \circ \mathbf{M}^{\text{intra}}, \tag{19}$$

where $\circ$ denotes element-wise multiplication. Next we get the average inter-class distance $AD^{\text{inter}}$ and the average intra-class distance $AD^{\text{intra}}$, with which we measure the over-smoothness as their ratio $R$:

$$AD^{\text{inter}} = \frac{\sum_{i,j} d_{ij}^{inter}}{\sum_{i,j} \mathbb{1}(d_{ij}^{inter})}, \tag{20}$$

$$AD^{\text{intra}} = \frac{\sum_{i,j} d_{ij}^{intra}}{\sum_{i,j} \mathbb{1}(d_{ij}^{intra})}, \tag{21}$$

$$R = \frac{AD^{\text{inter}}}{AD^{\text{intra}}}, \tag{22}$$

where $\mathbb{1}(x) = 1$ if $x > 0$ otherwise 0.

## E    FURTHER EXPERIMENTAL RESULTS

### E.1    CLASSIFICATION RESULTS UNDER THE LABEL-SCARCE SETTING

Supplementary experiments for Table 2 are shown in Table 6.

Table 6: Average test accuracy (%) and over-smoothness under the label-scarce setting (10 labels per class) with varying layers. For both metrics, the larger the better. A: Accuracy. S: Over-smoothness.

| Dataset | Method | 2 layers | 4 layers | 6 layers | 8 layers | 10 layers |
|---------|--------|----------|----------|----------|----------|-----------|
| Pubmed | Vanilla GCN [A / S] | 70.5 / 2.06 | 70.1 / 2.00 | 70.2 / 1.94 | 69.3 / 1.82 | 67.9 / 1.75 |
|  | **VEM-GCN** [A / S] | **72.6 / 2.14** | **72.5 / 2.10** | **72.5 / 2.07** | **72.6 / 2.12** | **72.4 / 2.08** |
| Amazon Photo | Vanilla GCN [A / S] | 89.5 / 4.04 | 83.5 / 3.87 | 80.4 / 3.21 | 78.5 / 3.10 | 54.8 / 2.88 |
|  | **VEM-GCN** [A / S] | **90.8 / 5.41** | **90.4 / 4.52** | **90.5 / 4.89** | **90.2 / 4.67** | **90.1 / 4.46** |
| Amazon Computers | Vanilla GCN [A / S] | 80.2 / 2.68 | 76.8 / 4.15 | 71.2 / 2.89 | 60.6 / 3.55 | 50.2 / 2.71 |
|  | **VEM-GCN** [A / S] | **82.6 / 4.18** | **82.2 / 4.59** | **82.4 / 4.31** | **82.6 / 4.20** | **82.1 / 3.99** |
| MS Academic | Vanilla GCN [A / S] | 89.6 / 7.77 | 84.3 / 6.67 | 82.1 / 6.61 | 80.1 / 6.35 | 73.7 / 6.28 |
|  | **VEM-GCN** [A / S] | **91.2 / 11.7** | **90.9 / 11.3** | **91.1 / 11.9** | **91.1 / 11.3** | **91.0 / 11.1** |

### E.2    VISUALIZATION RESULTS

Figure 2 demonstrates the topology optimization results on the Cora-ML dataset under the full-supervised setting. For the lable-scarce setting, the visualization results are shown in Figure 3.

We further take Cora-ML as an example dataset to provide t-SNE (Maaten & Hinton, 2008) visualizations of the learned node embeddings (input node features of the last layer) extracted by the vanilla GCN and our proposed VEM-GCN with varying layers under the label-scarce setting (10 labeled nodes per class). The results are shown in Figures 4 and 5. As can be seen, VEM-GCN indeed generates more separable node embeddings for nodes in different classes (colors) for classification. In particular, a ten-layer vanilla GCN severely suffers from the over-smoothing issue where node features in different classes are over-mixed and thus indistinguishable, while our VEM-GCN architecture with a ten-layer GCN as the backbone still achieves comparable performance with a two-layer model.

### E.3    COMPLEXITY ANALYSIS

VEM-GCN is very flexible and general. There is no constraint on the design of the two neural networks $q_\phi$ and $p_\theta$. Therefore, the architecture can be combined with arbitrary GCN models and

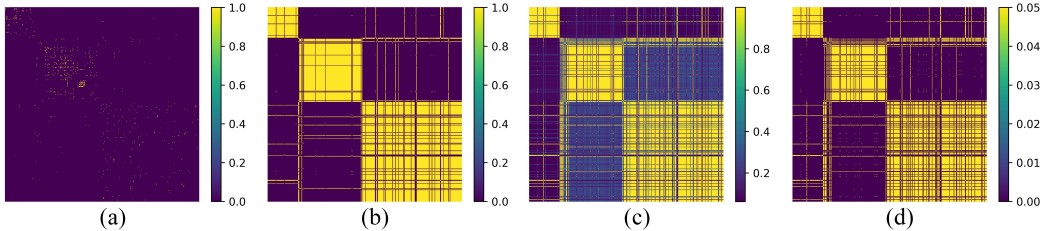

Figure 3: Visualization results of topology optimization on the Cora-ML dataset under the label-scarce setting (10 labels per class). We plot an induced subgraph (node indices from $450$ to $850$) for a better view. (a) The observed graph $\mathbf{A}_{\text{obs}}$; (b) The optimal graph $\tilde{\mathbf{A}}$; (c) The approximate posterior distribution $q_\phi$ on $\mathbf{A}_{\text{latent}}$; (d) The refined graph $\bar{q}_\phi$.

Table 7: Average test accuracy (%) of GCNII and VEM-GCNII on the three citation networks under the label-scarce setting (10 labels per class). The number in parentheses denotes the number of layers.

| Method | Cora | Citeseer | Pubmed |
|---|---|---|---|
| GCNII | 78.6 (64) | 64.3 (32) | 70.6 (16) |
| **VEM-GCNII** | **79.8** (64) | **66.2** (32) | **72.7** (16) |

node embedding methods. Moreover, we generalize some existing state-of-the-art strategies for tackling the over-smoothing problem (i.e., DropEdge and AdaEdge). In comparison to the vanilla GCN, VEM-GCN achieves these benefits with topology optimization at the cost of efficiency. As illustrated in Sections 3.2 and 3.3, the M-step is a traditional training procedure for optimizing the GCNs. Although $\mathbf{A}_{\text{latent}}$ recovers more intra-class edges than $\mathbf{A}_{\text{obs}}$, the parameter $p$ in Eq. 9 helps maintain the sparsity of the latent graph in each step of the training procedure. Thus the M-step shares almost the same complexity as GCN. The E-step introduces an extra procedure that trains a MLP for graph structure optimization. However, to address the over-smoothing issue at the core, we argue that topology optimization is necessary. Actually, efficiency issue is a common problem for some Bayesian approaches and topology optimization methods. Despite decreased efficiency compared with the vanilla GCN, VEM-GCN optimizes $\mathbf{A}_{\text{latent}}$ with amortized variational inference (i.e., training a MLP shared by all the node pairs with mini-batch SGD in the E-step), which is faster than BGCN (Zhang et al., 2019) (Bayesian method) and LDS (Franceschi et al., 2019) (topology optimization) and is scalable for large-scale graphs. For training on the Amazon Photo dataset with a NVIDIA GTX 1080 Ti GPU, VEM-GCN is about $3\times$ faster than LDS and $4\times$ faster than BGCN.

### E.4 VEM-GCNII

As mentioned above, our VEM-GCN architecture is flexible and general. In the E-step, the neural network can support arbitrary desirable node embeddings and the GCN in the M-step can be substituted with any graph models. This subsection further verifies the effectiveness of our proposed method by trying different models for $p_{\theta_1}(\mathbf{Y}_l|\mathbf{A}_{\text{latent}}, \mathcal{G}_{\text{obs}})$. We also apply the same node embeddings as illustrated in Appendix D. Trying more effective node embeddings is not the focus of this paper and is left for the future work.

Recently, Chen et al. (2020b) proposed a simple and deep GCN model termed GCNII to address the over-smoothing issue. GCNII improves the vanilla GCN via Initial residual and Identity mapping:

$$\mathbf{H}^{(l+1)} = \sigma\left(\left((1-\alpha^{(l)})\tilde{\mathbf{P}}\mathbf{H}^{(l)} + \alpha^{(l)}\mathbf{H}^{(0)}\right)\left((1-\beta^{(l)})\mathbf{I}_N + \beta^{(l)}\mathbf{W}^{(l)}\right)\right), \qquad (23)$$

where $\mathbf{H}^{(0)} = \mathbf{X}\mathbf{W}^{(0)}$ is the output of the first layer (a fully connected layer), $\tilde{\mathbf{P}} = \dot{\mathbf{D}}^{-\frac{1}{2}}\dot{\mathbf{A}}\dot{\mathbf{D}}^{-\frac{1}{2}}$ is the normalized adjacency matrix, $\alpha^{(l)}$ and $\beta^{(l)}$ are two hyperparameters.

We utilize GCNII as the backbone that results in the VEM-GCNII architecture. We use the official source code from https://github.com/chennnM/GCNII and employ the hyperparameter

settings reported in (Chen et al., 2020b) that achieve the best performance on the three citation networks (i.e., Cora, Citeseer, and Pubmed) under the label-scarce setting. For the other four datasets, we roughly tuned the hyperparameters but found that GCNII does not outperform the vanilla GCN. Thus, we only evaluate GCNII and VEM-GCNII on Cora, Citeseer, and Pubmed with 10 labeled nodes per class as the training set. Experimental results are shown in Table 7. As can be seen, VEM-GCNII consistently boosts the performance of GCNII, further verifying the effectiveness and flexibility of our proposed architecture.

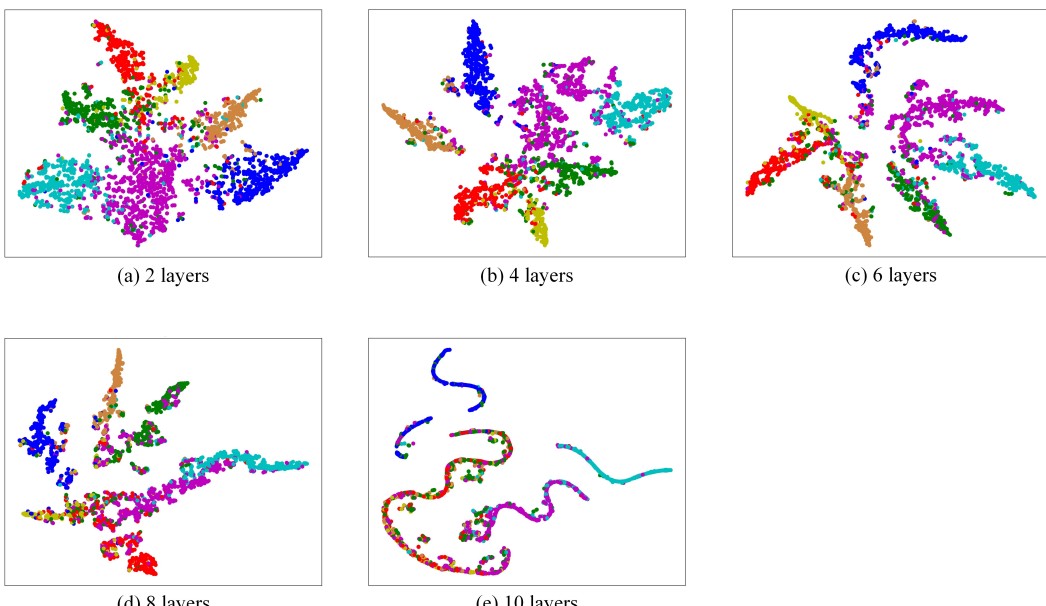

Figure 4: t-SNE plots of learned node embeddings extracted by vanilla GCN with varying layers on the Cora-ML dataset. Different colors denote different classes.

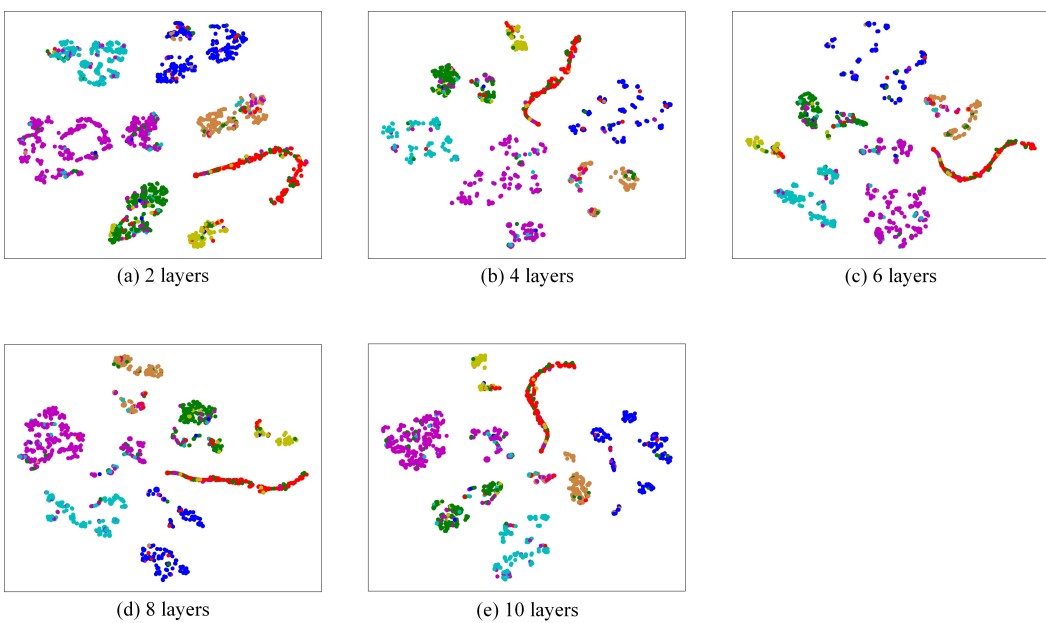

Figure 5: t-SNE plots of learned node embeddings extracted by VEM-GCN with varying layers on the Cora-ML dataset. Different colors denote different classes.

