# OpenReview forum: "VEM-GCN: Topology Optimization with Variational EM for Graph Convolutional Networks"
_ICLR.cc/2021/Conference — Reject_

### Official Review · AnonReviewer4 · 2020-10-28
**Review of VEM-GCN: Accept**

**Rating:** 8
**Confidence:** 4

**Review:**

Summary:

The authors present a method for tackling the problem of over-smoothing in graph convolutional networks. Specifically, this is achieved by explicitly modelling a latent graph which, ideally, would be a graph which connects an observation to all other observations of the same class and no observations of a different class. In practice, there is only an uncertain picture of this latent graph as in many applications the labels must be estimated for unlabelled observations. The authors present an EM variational algorithm for approximating both this latent graph and using it to improve the estimation of a GCN. The authors demonstrate that the proposed method performs favourably on a battery of test against an array of existing methods for solving the node classification problem.

Strengths:

The paper tackles an important question in the GCN literature, which is how to deal with situations in which the graph is unobserved or the observed graph structure is only a fraction of the true graph. The method proposed, modelling and optimising a latent graph, makes sense and is well justified.  The authors position the paper well in terms of its contributions in relation to previous work. This includes empirical comparisons to a wide array of existing competing methods. The paper is well written and clearly describes the proposed method.

Weaknesses:
The main weakness of the paper is in the empirical section. Specifically, I would like to seen an expanded Table 2 to include more comparisons with existing methods. From Table 1, I am convinced that VEM-GCN achieves similar performance to existing methods, even if the magnitude of the performance increase is not very large. However, I think the main contribution of the paper in the empirical section is Table 2 and should be the main focus. It clearly demonstrates that the proposed method reduces over-smoothing relative to a vanilla GCN as expected, especially as the depth of the model increases, but I wonder what a similar comparison to the other models in Table 1 would show. Could the authors provide the results of a similar analysis, at least for the models for which this is possible?

Reasons for score:

I vote for accepting the paper. It is a solid, clear, and novel contribution to the literature on GCNs that directly addresses an important consideration in these models that is often overlooked.

Questions for the rebuttal period:

Please refer to the questions in the weaknesses section.

---

> ### Author Response · Authors · 2020-11-21
> **Response to Reviewer 4**
>
> Thanks for your positive comments.
>
> Indeed, tackling the over-smoothing issue is the main contribution of this paper. According to your suggestion, we have revised Table 2 to consider two closely related methods for tackling over-smoothing (i.e., DropEdge and AdaEdge), and please kindly check it. DropEdge and AdaEdge can relieve the over-smoothing issue to some extent, but still suffers from performance degradation with deeper models. For PairNorm and BBGDC, the results in their papers show that the problem of performance degradation with increasing layers still cannot be well addressed (please refer to Figure 3 in PairNorm (Zhao & Akoglu 2020) and Figure 2 in BBGDC (Hasanzadeh et al., 2020)). These facts imply that VEM-GCN is more robust for addressing the over-smoothing problem.

---

### Official Review · AnonReviewer2 · 2020-10-29
**The paper introduces a method for alleviating the over-smoothing problem of GNNs, which is theoretically designed well but has some practical limitations.**

**Rating:** 6
**Confidence:** 5

**Review:**

This paper proposes a method to alleviate the over-smoothing problem of GNNs. The key idea is to generate a latent graph structure via leveraging stochastic block model to approximate the observed graph structure and label information. The learned latent graph is expected to have a clear community structure with dense intra-class edges and sparse inter-class edges, so that labels of unlabeled nodes are better predicted based on the latent structure. The whole framework is well designed as an MLE problem, with EBLO solved by an alternate EM style algorithm. Both E-step and M-step are assumed to enhance each other's performance, but this point is not clearly validated in the experiments. Also, it is good to see some discussions on the relationship between the proposed framework and dropedge and adaedge methods. Overall, the idea makes sense in terms of joint topology optimization (via SBM) and node classification. The methodology is designed well as an MLE problem. The paper writes well and the experimental results demonstrate effectiveness to some extent.

The following are some weak points regarding the paper.
1. The method is designed to only handle semi-supervised node classification problem, which may not be flexible enough to be associated with various GNNs for different downstream tasks.
2. The idea of joint topology optimization and node classification is not new. For example, the following work
Zheng et al. Robust Graph Representation Learning via Neural Sparsification. ICML 2020.
deals with a similar problem, and is flexible to different tasks. It is better to compare with it in the experiments. Thus the novelty of the proposed method lies in its incorporation of SBM and community optimization, which is good but not very substantial.
3. Because of the pairwise node link generation via SBM, the complexity of the method may be high. From the paper and the appendix, this point seems not discussed in detail. It is better to provide a clearer comparison between the complexity of the proposed method and some other recent topology optimization methods.
4. It seems the proposed method has many hyperparameters, such as p, epsilon in Eq. (9), S in Eq. (11). It is better to provide a summary about these parameters and how to select them in the experiments.
5. From experimental results table 1, the performance gain of the proposed method is around 1-2% accuracy compared to vanilla GCN. This seems not very significant, especially considering the practice whether it is necessary to cost more computations to achieve such a performance gain. Meanwhile, how much extra time to run the proposed method compared to pure GCN is not clear. Also, it is better to show variance to help understand the statistical significance.
6. From table 2, the accuracy improvement is slightly better than table 1 in the scarce-label setting. But table 3 tells if adding a small amount of labels, the difference between GCN and the proposed method becomes small again, which draws a similar question as toward the results in table 1.

---

> ### Author Response · Authors · 2020-11-21
> **Responses to Reviewer 2 [Part 1/2]**
>
> Thanks for your detailed comments. Below, we provide responses to your concerns.
>
> **C1: Problem setting**
>
> This paper proposes a novel architecture to address the over-smoothing issue. The over-smoothing issue is originally derived from the task of semi-supervised node classification and is a major concern for node classification with GCNs. Same as many existing methods for tackling over-smoothing (e.g. AdaEdge and PairNorm) and some other related works (e.g. LDS, GMNN, and the Bayesian methods), we only aim at solving semi-supervised node classification in this paper. Considering other graph-based tasks is not the focus of this paper.
>
> **C2: Novelty**
>
> NeuralSparse (Zheng et al. ICML 2020) also addresses node classification from the topological view. However, the motivation and methodology are quite different. NeuralSparse uses the Gumbel Softmax trick to sample $k$ neighbors from the observed neighborhood for each node. It indeed can remove some harmful edges (we view inter-class edges as harmful edges in this paper), but *it cannot recover missing helpful edges (intra-class edges)*. Moreover, besides the cross-entropy classification loss to optimize the model, there is no other supervised loss to tell how to select edges (i.e. *there is no explicit criterion for topology optimization in NeuralSparse*). By contrast, in the E-step of our VEM-GCN, we optimize $q_{\phi}$ to approximate SBM that explicitly enhances intra-class connection and suppresses inter-class interaction. Since NeuralSparse was published very recently and is not open source, we do not implement it due to the time limit. However, we perform an extra experiment to show that *only removing harmful edges from the original graph is not enough and adding more intra-class edges is also necessary*. We train the GCN with (1) the original graph, (2) the original graph with all ground-truth inter-class edges removed, and (3) the original graph with all ground-truth intra-class edges added. Experiments are performed on Cora with 10 labels per class:
>
> GCN (1): 74.5%
>
> GCN (2): 83.7%
>
> GCN (3): 100.0%
>
> We find that adding intra-class edges benefits the GCN more than removing inter-class edges. This fact implies that adding intra-class edges is necessary for performance improvement.
>
> Moreover, NeuralSparse can be combined with VEM-GCN since there is no constraint on the design of $p_{\theta_1}$ in the M-step.
>
> **C3: Complexity Analysis**
>
> The *M-step* of VEM-GCN is a traditional training procedure for optimizing the GCN. Although the optimized graph recovers many intra-class edges compared with the original graph, the parameter $p$ in $\bar{q}_{\phi}$ ($p<1$) helps maintain the sparsity of the adjacency matrix in the training procedure. Thus, the complexity of the M-step is almost the same as the GCN model.
>
> The *E-step* is to train a simple four-layer MLP with stochastic gradient descent to approximate SBM. We use amortized variational inference and thus we can sample a small batch size of graph nodes for each optimization step. Hence, VEM-GCN has lower complexity than existing topology optimization methods such as LDS. LDS needs to load all the $N$ graph nodes in memory for topology optimization and optimizes the graph with $\mathcal{O}(N^{2})$ parameters. LDS cannot scale to large graphs and run OOM on three large graphs (i.e. Pubmed, Amazon Computers, and MS Academic).

---

> > ### Author Response · Authors · 2020-11-21
> > **Responses to Reviewer 2 [Part 2/2]**
> >
> > **C4: Hyperparameter**
> >
> > VEM-GCN is not sensitive to $p$. We use $p<1$ to help maintain the sparsity of the latent graph in the training procedure and perform in a similar way as DropEdge. We use $p=1$ and $S=1$ (deterministic $\bar{q}_{\phi}$) to do inference once in the test procedure. The parameters $\varepsilon _1$ and $\varepsilon _2$ need careful tuning. We tune these two hyperparameters in the following search space: $\varepsilon_1 \in$ {0.95, 0.99, 0.995, 0.999, 0.9995, 0.9999} and $\varepsilon_2\in$ {0.001, 0.005, 0.01, 0.05, 0.1}. Detailed implementations and hyperparameter settings are shown in Appendix D.
> >
> > **C5: Efficiency**
> >
> > As illustrated in Appendix E.3, we agree that VEM-GCN achieves the benefits with topology optimization at the cost of efficiency. Compared with the vanilla GCN, the difference is that the E-step introduces an extra procedure that trains a four-layer MLP for graph structure optimization. We focus on topology optimization to address the over-smoothing issue in this paper and would consider to further improve the efficiency of topology optimization in future work. Actually, efficiency issue is a common problem for some Bayesian methods (e.g. BGCN) and topology optimization methods (e.g. LDS). As analyzed in Appendix E.3 and [C3: Complexity Analysis], VEM-GCN shows scalability for large graphs and is more efficient than some existing methods such as LDS and BGCN.
> >
> > **C6: Performance improvement**
> >
> > There might be no direct correlation between the performance improvement and the size of training data (e.g., small performance improvement for more training data). Taking Amazon Computers as an example, the accuracy improvement under the full-supervised setting is 3.3% (Table 1), while the accuracy improvement is 2.4% (Table 6 in Appendix E.1) for the label-scarce setting with 10 labels per class. Our model consistently boosts the performance of the backbone (GCN and GCNII in this paper) employed for $p_{\theta_1}$ in all cases, which is significant compared with a variety of baselines.
> >
> > For the question about the results in Table 3, firstly, we wish to point out that previous results with 30 labels per class had something wrong. The patience of the early stopping strategy was wrongly set, causing lower accuracy. We have reimplemented the corresponding results and please kindly check it. Secondly, the performance improvement indeed becomes small when we get more training data on Amazon Photo under the label-scarce setting. However, this is not always true on other datasets as illustrated above. We think the reason is that GCN itself can already achieve very high accuracy with a small amount of training data on Amazon Photo. When adding more training labels, the performance improvement of GCN itself becomes small. Since VEM-GCN uses GCN as the backbone, it also yields small gains for more training data.

---

### Official Review · AnonReviewer3 · 2020-10-29
**Novelty is limited**

**Rating:** 6
**Confidence:** 5

**Review:**

A brief summary of the paper.
This paper proposed a novel architecture termed VEM-GCN to address the over-smoothing problem of GNNs in the node classification task. The main idea is to optimize the graph topology by removing the inter-class edges as well as adding the intra-class edges, and then the noise information would not pass between nodes with different categories. The framework learns with two alternating steps: E-step optimizes the topology while M-step improves the performance of GCN. The experimental results show that the proposed VEM-GCN achieves higher classification accuracy than baseline methods.

Main contributions of the paper.
1.	This paper proposes a joint learning framework for GNN classification model and graph topology, which leverages variational EM as a learning framework.
2.	This paper presents a graph topology learning algorithm based on SBM and neural networks, which employs the node embedding and labels to optimize the topology.
3.	Extensive experiments to demonstrate the performance superior of the proposed method.

Strengths
The strengths of this paper are summarized below,
1.	The proposed method is clearly introduced. Concretely, the theoretical background and the algorithm details are both well defined and written in a clear way.
2.	The paper is well-organised and well-written.
3.	Extensive experiments are conducted and results analysis are given. The visualization result is intuitive to demonstrate the property of the proposed method.

Weaknesses
The weaknesses of this paper are summarized below,
1.	The main hesitation with this paper is the novelty of the proposed method. Actually, optimizing the graph topology is a hot research direction, and a variety of works are presented about this topic recently. In addition to the AdaEdge, LDS and TO-GCN mentioned in the paper, other works, e.g., "Graph-Revised Convolutional Network" (ECML-PKDD 2020, arxiv: 1911.07123), "Deep Iterative and Adaptive Learning for Graph Neural Networks" (arxiv: 1912.07832), and "Graph Structure Learning for Robust Graph Neural Networks" (arxiv: 2005.10203), also study the same problem. Among them, DIAL-GNN also leverages an iterative and alternating framework to learn GNN and topology, which is similar to this paper. The differences between LDS/AdaEdge and this work are also minor. GRCN and TO-GCN optimize topology and GNN simultaneously in an end-to-end way, which seems more efficient than this work. In summary, the novelty of the presented VEM-GNN is a little bit minor.
2.	A minor concern is about the motivation of this paper. The authors claim that "over-smoothing is caused by "indistinguishable features of nodes in different classes produced by the message passing along inter-class edges", so "adding intra-class edges and removing inter-class edges are helpful to suppress over-smoothing". However, I have a different understanding about the over-smoothing problem. A more common definition (by Li et al., 2018) about over-smoothing is "if a GCN is deep with many convolutional layers, the output features may be oversmoothed and vertices from different clusters may become indistinguishable." In my opinion, the over-smoothing is caused by the depth (or receptive field) of GNN and the message passing manner, but irrelevant to the graph topology. Assuming that we remove all the inter-class edges and connect all the intra-class edges in the graph. In such a situation, when the GNN goes deep, the output embeddings of each node with the same class still become indistinguishable (the embeddings of nodes in the same class will converge to an identical embedding). Maybe we can obtain a perfect classification model by this way, but the embeddings are still failed to represent the property of each node, and they are useless to be applied to other tasks (e.g., anomaly detection). In summary, I agree that topology optimization is beneficial to enhance the node classification performance, but its effect on surpassing over-smoothing is suspicious.
3.	The impact of each module/design in the proposed framework is not clearly stated. Specifically, a probability matrix $\bar{q}_{\phi}$ is acquired by the learned adjacency matrix <${q}_{\phi}$>, and then the adjacency matrix for GCN is sampled by <$\bar{q}_{\phi}$>. The question is: why don't we define $p=1$ directly to obtain a definite adjacency matrix? Such a design can be viewed as a "DropEdge", so is that the main contribution term for restraining over-smoothing? Authors should add more ablation study to demonstrate the impact of "learned topology" and "probability matrix" respectively.

Correctness and Clarity
The claims and method are well written without significant errors. The paper is well-organised and written in a logical way.

Additional Comments
Here are some additional comments for the authors,
1.	More baselines considering topology optimization should be included, such as GRCN, DIAL-GNN, Pro-GNN (the papers of these methods are mentioned in Weaknesses Section).
2.	It is better to demonstrate how much edges are added/removed since sparsity is an important factor affecting the efficiency of GCN.

---

> ### Author Response · Authors · 2020-11-21
> **Responses to Reviewer 3 [Part 1/2]**
>
> Thanks for the detailed comments. We would like to first clarify your concern (Concern2) about the over-smoothing issue, and subsequently, elaborate the differences between our work and the recent papers mentioned in Concern1. We also illustrate the setting of $\bar{q}_{\phi}$ and have added more baselines as suggested. Please kindly check it.
>
> **C2: Over-smoothing**
>
> First of all, we wish to point out that over-smoothing is concerned specially to measure node features in *different* classes in the task of *semi-supervised node classification*. As in (Li et al. AAAI 2018), “the graph convolution of the GCN model is actually a special form of Laplacian smoothing”. Therefore, when a GCN is deep, “the features of vertices within each connected component of the graph will converge to the same values”. This fact leads to over-smoothing that means nodes from different clusters (classes) become indistinguishable. As a result, over-smoothing is relevant to the connectivity of nodes (graph topology). As for nodes in the same class, Laplacian smoothing makes their features similar, which makes the subsequent classification task much easier.
>
> Moreover, existing literature also shows that *over-smoothing is relevant to topology*. Chen et al. (AAAI 2020a) first measured and relieved over-smoothing from the topological view. They proved that intra-class edges pass useful information to help classify nodes in the same class (that have similar features), whereas inter-class edges pass harmful noise that over-mixes node features in different classes. The reason for the performance degradation with deeper models is that a larger receptive field would contain more inter-class edges that bring in more noise. Based on this consideration, AdaEdge was developed to add intra-class edges and remove inter-class edges in a simple self-training way. However, AdaEdge still suffers from the problem of wrong graph adjustment operation (please see the Conclusion in (Chen et al. 2020a) and Section 3.4 in our paper). By contrast, our VEM-GCN enhances intra-class connection and suppresses inter-class interaction in a more principled way based on variational EM and empirically achieves better performance.
>
> **C1: Novelty**
>
> Although many works have studied topology optimization for GCNs, the motivations and methodology are quite different from our method. Few of them explicitly enhance intra-class connection and suppress inter-class interaction to **address the over-smoothing issue, which is the main contribution of this paper**. AdaEdge adds intra-class edges and removes inter-class edges in a simple self-training way, but still suffers from the problem of wrong graph adjustment operation. We elaborate the differences as follows.
>
> C1.1 GRCN (PKDD 2020) and DIAL-GNN (NeurIPS 2020)
>
> GRCN modifies the original adjacency matrix by adding a residual adjacency matrix in which each element is the dot product of the corresponding node embeddings generated by another GCN. DIAL-GNN (its newest version termed IDGL was published in NeurIPS 2020 during the review process) modifies the original adjacency matrix similar to GRCN. The optimized adjacency matrix is a linear combination of the original adjacency matrix and a matrix that measures the similarity between the node features. The graph learning metrics of IDGL are only adopted as regularization terms to make the graph connected and sparse. *Both GRCN and IDGL do not consider the over-smoothing issue*. They update the graph topology by considering the similarity matrices of nodes, rather than explicitly adjust the graph (i.e., enhance intra-class connection and suppress inter-class interaction) for better distinguishing the nodes. However, the similarity matrices in GRCN and IDGL are computed via the output node embeddings of GCN. If the GCN suffers from the over-smoothing issue, the computed similarity matrices would have large deviation from the actual ones and cause wrong graph adjustment operation (e.g., connect inter-class nodes). In comparison, our VEM-GCN aims at addressing the over-smoothing issue. The E-step approximates the assortative-constrained SBM to explicitly enhance intra-class connection and suppress inter-class interaction. As a result, the motivations and methodology of our work are quite different from these two papers. We have compared with GRCN and IDGL using their official codes in Tables 1 and 3 in the revised version.

---

> > ### Author Response · Authors · 2020-11-21
> > **Responses to Reviewer 3 [Part 2/2]**
> >
> > C1.2 Pro-GNN (KDD 2020)
> >
> > Pro-GNN aims at *defending adversarial attacks*, which is totally different from VEM-GCN. It empirically finds that adversarial edges damage the low rank and sparsity properties of real-world graphs. Thus, Pro-GNN introduces these two terms to the loss function as graph regularization and optimizes the regularized loss function. In comparison, our method does not introduce these constraints to the latent graph. Our motivation is to address the over-smoothing issue by enhancing intra-class connection and suppressing inter-class interaction. Under the setting for semi-supervised learning adopted in this paper, Pro-GNN does not obtain state-of-the-art performance as in defending adversarial attacks and only achieves similar performance as GCN (please refer to (Jin et al., 2020)). As a result, we do not include Pro-GNN in experimental evaluations.
> >
> > C1.3 AdaEdge (AAAI 2020)
> >
> > As illustrated in Section 3.4, AdaEdge adjusts the graph topology in a simple self-training manner. Although designed to enhance intra-class connection and suppress inter-class interaction, AdaEdge depends on (and would also be affected by) the prediction outcomes of the GCN, especially when the GCN suffers from over-smoothing. AdaEdge adjusts the edges connected by nodes that have already been classified by the GCN with high confidence. Thus, the performance improvement is limited by the output of the GCN and would even get worse for some misclassified nodes. For example, if a node is wrongly classified with high confidence, AdaEdge would directly connect this node with the community that it should not belong to, and introduce more harmful inter-class edges. On the contrary, our VEM-GCN directly optimizes the posterior distribution of the latent graph to approximate the assortative-constrained SBM. Therefore, VEM-GCN explicitly enhances intra-class connection and suppresses inter-class interaction to improve the performance.
> >
> > C1.4 LDS (ICML 2019) and TO-GCN (IJCAI 2019)
> >
> > LDS and TO-GCN also focus on topology optimization, rather than addressing the over-smoothing issue. As illustrated in our paper, LDS optimizes graph topology by solving a bilevel programming, which is quite different from VEM-GCN using variational EM. LDS uses the validation set for training and cannot scale to large graphs since it parameterizes the optimized graph of $N$ nodes with $\mathcal{O}(N^{2})$ parameters. Moreover, the $\mathcal{O}(N^{2})$ parameters are learned based on the validation loss of GCNs. This fact implies that learning performance would become worse, when the GCN suffers from the over-smoothing issue. TO-GCN only adds intra-class edges derived from the labeled nodes but does not consider improving the topology of unlabeled nodes, which causes topology imbalance between the labeled nodes and unlabeled nodes. The GCN trained on the labeled nodes with enhanced graph topology would fail on the unlabeled nodes with the original graph topology.
> >
> > Moreover, all the methods above can be combined with VEM-GCN since there is no constraint on the design of $p_{\theta_1}$ in the M-step. These aforementioned methods do not contradict our motivation to explicitly enhance intra-class connection and suppress inter-class connection.
> >
> > **C3: Probability matrix of the latent graph**
> >
> > VEM-GCN is not sensitive to $p$ in $\bar{q}_{\phi}$. The reason for not setting $p=1$ in the training procedure is that this causes a dense matrix which is inefficient for training (i.e. nodes may recover many intra-class edges). In the test procedure, we set $p=1$ to do inference once. Setting $p<1$ in the training procedure is to maintain the sparsity of the latent graph for efficient training while perform similar as DropEdge. Using $p=1$ also performs well but is inefficient. We give the visualization results of topology optimization in Figure 2.
> >
> > Taking Cora as an example dataset, we give the ablation study on $p$ under the label-scarce setting (10 labels per class in Table 2). The results are average test accuracy over 5 random data splits with 10 runs in each split.
> >
> > GCN (baseline): 74.52%
> >
> > VEM-GCN (p=1.0): 77.72%
> >
> > VEM-GCN (p=0.8): 77.69%
> >
> > VEM-GCN (p=0.6): 77.74%
> >
> > VEM-GCN (p=0.4): 77.68%
> >
> > VEM-GCN (p=0.2): 77.22%

---

### Official Review · AnonReviewer1 · 2020-10-29
**Jointly learning graph and labels**

**Rating:** 6
**Confidence:** 4

**Review:**


In this work the authors start from the following basic observation, that I will state in terms of binary classification.  In an ideal setting where there exist two labels, the graph structure should be two distinct connected components, and according to the author(s) the most natural choice is that the each component is a clique. However, when one performs semi-supervised learning, edges going across communities over smooth the labels, and especially in the absence of many labeled points this causes big issues in node classification. For this reason the authors assume that the graph is actually a noisy version of some latent graph. They incorporate a variational approach to GCNs, as a novel architecture that iteratively refines the node labels and the graph. Figure 2 nicely summarizes how  the proposed architecture enhances the community structure. Some comments to the authors follow.


- Could the authors discuss alternative choices to l2 minimization such as the paper Algorithms for Lipschitz Learning on Graphs by Rasmus Kyng et al. at uses Lp-norm minimization as means to avoid over smoothing in a different but super relevant context of label propagation?
- The authors discuss drop edge. While this is an issue of the DropEdge method, wouldn't it make sense to remove edges that have higher effective resistance instead of random sampling? The latter is more likely to kill intra-edges, instead of the inter-edges that cause the oversmoothing? It would be nice to include such a baseline in the experiments.
- What happens when instead of having in the ideal setting two connected components that are cliques, there are two bipartite cliques instead?  This would capture a notion of 'heterophily' instead of 'homophile' that naturally creates a clique. Can your method be extended to this case?
- Can you prove that when the input graph is stochastic block model your method provably results in the right classification? It seems such a claim could be plausible to prove analytically, at least when the gap of intra- vs inter-community edges is large enough.

---

> ### Author Response · Authors · 2020-11-21
> **Responses to Reviewer 1 [Part 1/2]**
>
> Thanks for your detailed comments. Below, we provide our responses.
>
> **C1: $p$-Laplacian minimization**
>
> Label propagation (LP) is a fast algorithm that uses network topology alone to detect the communities in a graph. Standard Laplacian regularization algorithm (corresponding to a vector space setting with $2$-norm) might induce constant label function almost everywhere with extremely thin spikes. The $p$-Laplacian minimization for large $p$ interpolates smoothly over the graph and does not exhibit undesirable “flatness” behavior in the graph. Kyng et al. (the paper entitled “Algorithms for Lipschitz Learning on Graphs”) compute the absolutely minimal Lipschitz extension to assign labels as smooth as possible across edges. In this paper, we study Graph Convolutional Networks (GCNs), which is different from LP in the sense of objective function, definition of “smooth”, and input graph data.
>
> C1.1 Objective function
>
> In solving semi-supervised node classification with GCNs (the M-step in our work), the objective function is to maximize the log-likelihood of the observed node labels or to minimize the cross-entropy classification loss. According to (Kipf & Welling, 2017), GCN directly uses a neural network model to encode the graph structure and do not explicitly introduce graph-based regularization in the loss function.
>
> C1.2 Defintion of “smooth”
>
> The “smooth” in $p$-Laplacian minimization problem means that the node label values are locally smoothed (i.e., to learn the smoothest extension of the given node labels across the graph topology), while the “smooth” in GCN is to learn smooth node features for nodes in the same class. In GCN, the intra-class edges pass useful information to produce smooth features for nodes in the same class and the inter-class edges pass harmful noise to over-mix features of nodes in different classes. Thus, we propose to generate a topology that enhances intra-class connection and suppresses inter-class interaction of the observed graph.
>
> C1.3 Input graph data
>
> $p$-Laplacian minimization problem is solved using only the graph topology, while GCN leverages both graph topology and node attributes to learn node representations for classification.
>
> *Although the over-smoothing issues in GCNs and LP are different, we find that some works have tried to unify GCNs and LP (e.g., [1]). It would be possible to extend it to address the over-smoothing issue in GCNs in the future.*
>
> **C2: Removing inter-class edges**
>
> As suggested, we have added such baseline termed DropICE in Table 1 (full-supervised setting) and please kindly check it. DropICE removes the inter-class edges derived from the labeled nodes. Under the full-supervised setting, Table 1 shows that DropICE is inferior to DropEdge and the vanilla GCN. The reason for explaining the results is similar to TO-GCN. Only adding intra-class edges (TO-GCN) or removing inter-class edges (DropICE) for the labeled nodes might cause *topology imbalance* between the labeled nodes and unlabeled nodes as the topology of unlabeled nodes is not considered. The GCN trained on the labeled nodes with enhanced graph topology would fail on the unlabeled nodes with the original graph topology. Under the label-scarce setting, DropICE is similar to the vanilla GCN, as the number of the removed inter-class edges derived from the labeled nodes is small. Therefore, we do not include the results of DropICE in Table 2.

---

> > ### Author Response · Authors · 2020-11-21
> > **Responses to Reviewer 1 [Part 2/2]**
> >
> > **C3: Tackling graphs with heterophily**
> >
> > Most existing GCNs are based on the assumption of homophily and most common datasets (the seven benchmarks in our paper) adhere to this principle [2]. Many popular GCNs fail to generalize to the heterophily setting and are even worse than the models that ignore the graph structure (e.g., Multi-Layer Perceptron) [3]. Designing novel architectures for tackling graphs with heterophily is another hot research direction but not the focus of this paper.
> > Our work is proposed by considering the over-smoothing problem from the topological perspective. [4] proves that the performance of GCN is highly correlated with the homophily of the graph. The graph with more intra-class edges and fewer inter-class edges suggests lower risk of over-mixing node features in different classes, and consequently, leads to better performance in node classification with GCNs. Based on this consideration, we introduce a latent graph to enhance intra-class connection and suppress inter-class interaction of the observed graph.
> >
> > **C4: Proof**
> >
> > We do not prove that, when the input graph is a stochastic block model (SBM), our method results in the right classification. We provide some discussions as follows.
> >
> > C4.1 In the disassortative-constrained SBM (i.e. heterophily), the performance of GCN would not be good. Please refer to our response in [C3: Tackling graphs with heterophily].
> >
> > C4.2 Our method has no assumption on the generation of input graphs. Our motivation is to generate a latent graph that has better homophily than the observed graph (i.e. to enhance intra-class connection and suppress inter-class interaction), which empirically leads to better classification performance with GCNs as studied in previous studies [4].
> >
> > C4.3 The performance of GCN is correlated with both graph topology and input node attributes. As studied in [5], a graph with a clear topology generated by a strongly assortative SBM but random node features would not achieve very high accuracy. However, given the input node attributes, the clearer the topology is, the better performance a GCN achieves [4]. In my humble opinion, GCNs are not guaranteed to result in the right classification in any SBM, so does VEM-GCN.
> >
> > == References ==
> >
> > [1] Unifying Graph Convolutional Neural Networks and Label Propagation. Under Review at ICLR2021. https://openreview.net/forum?id=oh71uL93yay
> >
> > [2] Klicpera et al. Diffusion Improves Graph Learning. In NeurIPS 2019.
> >
> > [3] Zhu et al. Beyond Homophily in Graph Neural Networks: Current Limitations and Effective Designs. In NeurIPS 2020.
> >
> > [4] Chen et al. Measuring and Relieving the Over-smoothing Problem for Graph Neural Networks from the Topological View. In AAAI 2020.
> >
> > [5] Wang et al. AM-GCN: Adaptive Multi-channel Graph Convolutional Networks. In KDD 2020.

---

### Author Response · Authors · 2020-11-21
**Overall Responses to All the Reviewers**

We sincerely thank all the reviewers for the constructive comments to help us improve the paper! Overall, the reviewers give a positive evaluation on our work to address the over-smoothing issue in GCNs. We hope to answer all questions and provide clarifications in responses to each reviewer. We have uploaded a rebuttal version of our paper and made the following revisions.

(1) According to the suggestions from Reviewers 2 and 3, we have discussed more methods for topology optimization in the section of Related Works. Further experimental results are provided in Tables 1 and 3 in the revised version. For more discussions please refer to the responses to Reviewer 2 [C2] and Reviewer 3 [C1].

(2) As suggested by Reviewer 4, we have extended Table 2 to include two closely related methods for tackling over-smoothing (i.e., AdaEdge and DropEdge).

(3) Following the suggestion from Reviewer 1, we have considered a new baseline that removes all inter-class edges derived from the labeled nodes, which is named as DropICE. Experimental results are shown in Table 1 in the revised version.

(4) Thanks for the question of Reviewer 2 about the results in Table 3. We find that previous results on Amazon Photo with 30 labels per class have something wrong. The patience of the early stopping strategy was wrongly set, thus causing lower accuracy. We have reimplemented the corresponding experiments and revised the results. Please kindly check it!

(5) We provide the detailed implementations (e.g., the setting of $\bar{q}_{\phi}$) and the hyperparameter settings in Appendix D. For complexity analysis, please refer to Appendix E.3 and our response to Reviewer 2 [C3].

---

### Decision · Program_Chairs · 2021-01-07
**Final Decision**

**Decision:**

Reject

**Comment:**

The authors propose a new approach to topology optimization to address over-smoothing in GCNs. This is a borderline paper. Topology optimization is clearly important and relevant and the approach tries to optimize the topology (add/delete edges) by viewing the problem as a latent variable model and aiming to optimize the graph together with the GCN parameters to maximize the likelihood of observed node labels. A number of related joint topology optimization approaches exist, however, as discussed in the reviews and the responses. The proposed methodology is termed variational EM but is a bit heuristic in the sense that E and M steps do not follow a consistent criterion (the direction of KL is flipped between the steps). A number of comparisons are provided with consistent gains though the gains appear relatively small. No error bars are provided despite request to add them to better assess the significance of these results. It remains unclear whether the gains are worth the added complexity.